# ON ACHIEVING OPTIMAL ADVERSARIAL TEST ERROR

**Justin D. Li & Matus Telgarsky**
University of Illinois, Urbana-Champaign
{jdli3,mjt}@illinois.edu

## ABSTRACT

We first elucidate various fundamental properties of optimal adversarial predictors: the structure of optimal adversarial convex predictors in terms of optimal adversarial zero-one predictors, bounds relating the adversarial convex loss to the adversarial zero-one loss, and the fact that continuous predictors can get arbitrarily close to the optimal adversarial error for both convex and zero-one losses. Applying these results along with new Rademacher complexity bounds for adversarial training near initialization, we prove that for general data distributions and perturbation sets, adversarial training on shallow networks with early stopping and an idealized optimal adversary is able to achieve optimal adversarial test error. By contrast, prior theoretical work either considered specialized data distributions or only provided training error guarantees.

## 1 INTRODUCTION

Imperceptibly altering the input data in a malicious fashion can dramatically decrease the accuracy of neural networks (Szegedy et al., 2014). To defend against such *adversarial attacks*, maliciously altered training examples can be incorporated into the training process, encouraging robustness in the final neural network. Differing types of attacks used during this adversarial training, such as FGSM (Goodfellow et al., 2015), PGD (Madry et al., 2019), and the C&W attack (Carlini & Wagner, 2016), which are optimization-based procedures that try to find bad perturbations around the inputs, have been shown to help with robustness. While many other defenses have been proposed (Guo et al., 2017; Dhillon et al., 2018; Xie et al., 2017), adversarial training is the standard approach (Athalye et al., 2018). Despite many advances, a large gap still persists between the accuracies we are able to achieve on non-adversarial and adversarial test sets. For instance, in Madry et al. (2019), a wide ResNet model was able to achieve 95% accuracy on CIFAR-10 with standard training, but only 46% accuracy on CIFAR-10 images with perturbations arising from PGD bounded by $8/255$ in each coordinate, even with the benefit of adversarial training.

In this work we seek to better understand the optimal adversarial predictors we are trying to achieve, as well as how adversarial training can help us get there. While several recent works have analyzed properties of optimal adversarial zero-one classifiers (Bhagoji et al., 2019; Pydi & Jog, 2020; Awasthi et al., 2021b), in the present work we build off of these analyses to characterize optimal adversarial convex surrogate loss classifiers. Even though some prior works have suggested shifting away from the use of convex losses in the adversarial setting because they are not adversarially calibrated (Bao et al., 2020; Awasthi et al., 2021a;c; 2022a;b), we show the use of convex losses is not an issue as long as a threshold is appropriately chosen.

We will also show that under idealized settings adversarial training can achieve the optimal adversarial test error. In prior work guarantees on the adversarial test error have been elusive, except in the specialized case of linear regression (Donhauser et al., 2021; Javanmard et al., 2020; Hassani & Javanmard, 2022). Our analysis is in the Neural Tangent Kernel (NTK) or near-initialization regime, where recent work has shown analyzing gradient descent can be more tractable (Jacot et al., 2018; Du et al., 2018). Of many such works our analysis is closest to Ji et al. (2021), which provides a general test error analysis, but for standard (non-adversarial) training.

A recent work (Rice et al., 2020) suggests that early stopping helps with adversarial training, as otherwise the network enters a robust overfitting phase in which the adversarial test error quickly rises while the adversarial training error continues to decrease. The present work uses a form of early stopping, and so is in the earlier regime where there is little to no overfitting.

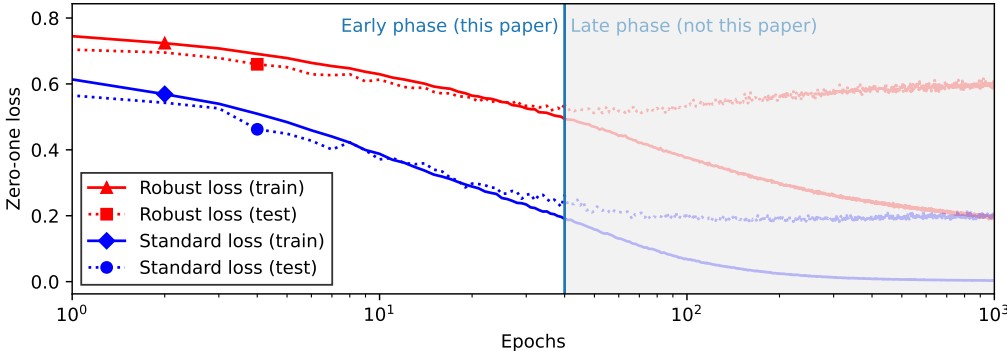

Figure 1: A plot of the (robust/standard) zero-one (training/test) loss throughout training for an adversarially trained network. We ran Rice et al.'s code, using a constant step size of $0.01$. The present work is set within the early phase of training, where we can get arbitrarily close to the optimal adversarial test error. In fact, due to technical reasons our analysis will be further restricted to an even earlier portion of this phase, as we remain within the near-initialization/NTK regime. As noted in prior work, adversarial training, as compared with standard training, seems to have more fragile test-time performance and quickly enters a phase of severe overfitting, but we do not consider this issue here.

## 1.1 OUR CONTRIBUTIONS

In this work, we prove structural results on the nature of predictors that are close to, or even achieve, optimal adversarial test error. In addition, we prove adversarial training on shallow ReLU networks can get arbitrarily close to the optimal adversarial test error over all measurable functions. This theoretical guarantee requires the use of optimal adversarial attacks during training, meaning we have access to an oracle that gives, within an allowed set of perturbations, the data point which maximizes the loss. We also use early stopping so that we remain in the near-initialization regime and ensure low model complexity. The main technical contributions are as follows.

1. **Optimal adversarial predictor structure (Section 3).** We prove fundamental results about optimal adversarial predictors by relating the global adversarial convex loss to global adversarial zero-one losses (cf. Lemma 3.1). We show that optimal adversarial convex loss predictors are directly related to optimal adversarial zero-one loss predictors (cf. Lemma 3.2). In addition, for predictors whose adversarial convex loss is almost optimal, we show that when an appropriate threshold is chosen its adversarial zero-one loss is also almost optimal (cf. Theorem 3.3). This theorem translates bounds on adversarial convex losses, such as those in Section 4, into bounds on adversarial zero-one losses when optimal thresholds are chosen. Using our structural results of optimal adversarial predictors, we prove that continuous functions can get arbitrarily close to the optimal test error given by measurable functions (cf. Lemma 3.4).

2. **Adversarial training (Section 4).** Under idealized settings, we show adversarial training leads to optimal adversarial predictors.

   (a) **Generalization bound.** We prove a near-initialization generalization bound for adversarial risk (cf. Lemma 4.4). To do so, we provide a Rademacher complexity bound for linearized functions around initialization (cf. Lemma 4.5). The overall bound scales directly with the parameter's distance from initialization, and $1/\sqrt{n}$, where $n$ is the number of training points. Included in the bound is a perturbation term which depends on the width of the network, and in the worst case scales like $\tau^{1/4}$, where $\tau$ bounds the $\ell_2$ norm of the perturbations.

   (b) **Optimization bound.** We show that using an optimal adversarial attack during gradient descent training results in a network which is adversarially robust on the training set, in the sense that it is not much worse compared to an arbitrary reference network (cf. Lemma 4.6). Comparing to a reference network instead of just ensuring low training error (as in prior work) will be key to obtaining a good generalization analysis, as the optimal adversarial test error may be high.

(c) **Optimal test error.** As the generalization and optimization bounds are both in a near-initialization setting, these two bounds can be used in conjunction. We first bound the test error of our trained network in terms of its training error using the generalization bound, and then apply the optimization bound to compare against training error of an arbitrary reference network. Another application of our generalization bound then allows us to compare against the test error of an arbitrary reference network (cf. Theorem 4.1). Applying approximation bounds and Lemma 3.4 then lets us bound our trained network's test error in terms of the optimal test error over all measurable functions (cf. Corollary 4.2).

## 2 RELATED WORK

We highlight several papers in the adversarial and near-initialization communities that are relevant to this work.

**Optimal adversarial predictors.** Several works study the properties of optimal adversarial predictors when considering the zero-one loss (Bhagoji et al., 2019; Pydi & Jog, 2020; Awasthi et al., 2021b). In this work, we are able to understand optimal adversarial predictors under convex losses in terms of those under zero-one losses, although we will not make use of any properties of optimal zero-one adversarial predictors other than the fact that they exist. Other works study the inherent tradeoff between robust and standard accuracy (Tsipras et al., 2019; Zhang et al., 2019), but these are complementary to this work as we only focus on the adversarial setting.

**Convex losses.** Several works explore the relationship between convex losses and zero-one losses in the non-adversarial setting (Zhang, 2004; Bartlett et al., 2006). Whereas the optimal predictor in the non-adversarial setting can be understood locally at individual points in the input domain, it is difficult to do so in the adversarial setting due to the possibility of overlapping perturbation sets. As a result, our analysis will be focused on the global structure of optimal adversarial predictors. Convex losses as an integral over reweighted zero-one losses have appeared before (Savage, 1971; Schervish, 1989; Hernández-Orallo et al., 2012), and we will adapt and make use of this representation in the adversarial setting.

**Adversarial surrogate losses.** Several works have suggested convex losses are inappropriate in the adversarial setting because they are not calibrated, and instead propose using non-convex surrogate losses (Bao et al., 2020; Awasthi et al., 2021a;c; 2022a;b). In this work, we show that with appropriate thresholding convex losses are calibrated, and so are an appropriate choice for the adversarial setting.

**Near-initialization.** Several works utilize the properties of networks in the near-initialization regime to obtain bounds on the test error when using gradient descent (Li & Liang, 2018; Arora et al., 2019; Cao & Gu, 2019; Nitanda et al., 2020; Ji & Telgarsky, 2019; Chen et al., 2019; Ji et al., 2021). In particular, this paper most directly builds upon the prior work of Ji et al. (2021), which showed that shallow neural networks could learn to predict arbitrarily well. We adapt their analysis to the adversarial setting.

**Adversarial training techniques.** Adversarial training initially used FGSM (Goodfellow et al., 2015) to find adversarial examples. Numerous improvements have since been proposed, such as iterated FGSM (Kurakin et al., 2016) and PGD (Madry et al., 2019), which strives to find even stronger adversarial examples. These works are complementary to ours, because here we assume that we have an optimal adversarial attack, and show that with such an algorithm we can get optimal adversarial test error. Some of these alterations (Zhang et al., 2019; Wang et al., 2021; Miyato et al., 2018; Kannan et al., 2018) do not strictly attempt to find a maximal adversarial attack at every iteration, but instead use some other criteria. However, Rice et al. (2020) proposes that many of the advancements to adversarial training since PGD can be matched with early stopping. Our work corroborates the power of the early stopping in adversarial training as we use it in our analysis.

**Adversarial training error bounds.** Several works are able to show convergence of the adversarial *training* error. Gao et al. (2019) did so for networks with smooth activations, but is unable to handle constant-sized perturbations as the width increases. Meanwhile Zhang et al. (2020) uses ReLU

activations, but imposes a strong separability condition on the training data. Our training error bounds use ReLU activations, and in contrast to these previous works simultaneously hold for constant-sized perturbations and consider general data distributions. However, we note that the ultimate goals of these works differ, as we focus on adversarial *test* error.

**Adversarial generalization bounds.** There are several works providing adversarial generalization bounds. They are not tailored to the near-initialization setting, and so they are either looser or require assumptions that are not satisfied here. These other approaches include SDP relaxation based bounds (Yin et al., 2018), tree transforms (Khim & Loh, 2019), and covering arguments (Tu et al., 2019; Awasthi et al., 2020; Balda et al., 2019). Our generalization bound also uses a covering argument, but pairs this with a near-initialization decoupling. There are a few works that are able to achieve adversarial test error bounds in specialized cases. They have been analyzed when the data distribution is linear, both when the model is linear too (Donhauser et al., 2021; Javanmard et al., 2020), and for random features (Hassani & Javanmard, 2022). In the present work, we are able to handle general data distributions.

## 3 PROPERTIES OF OPTIMAL ADVERSARIAL PREDICTORS

This section builds towards Theorem 3.3, relating zero-one losses to convex surrogate losses.

### 3.1 SETTING

We consider a distribution $\mathcal{D}$ with associated measure $\mu$ that is Borel measurable over $X \times Y$, where $X \subseteq \mathbb{R}^d$ is compact, and $Y = \{-1, 1\}$. For simplicity, throughout we will take $X = B_1$ to be the closed Euclidean ball of radius 1 centered at the origin. We allow arbitrary $P(y = 1|x) \in [0, 1]$ — that is, the true labels may be noisy.

We will consider general adversarial perturbations. For $x \in B_1$, let $\mathcal{P}(x)$ be the closed set of allowed perturbations. That is, an adversarial attack is allowed to change the input $x$ to any $x' \in \mathcal{P}(x)$. We will impose the natural restrictions that $\emptyset \neq \mathcal{P}(x) \subseteq B_1$ for all $x \in B_1$. That is, there always exists at least one perturbed input, and perturbations cannot exceed the natural domain of the problem. In addition, we will assume the set-valued function $\mathcal{P}$ is upper hemicontinuous. That is, for any $x \in X$ and any open set $U \supseteq \mathcal{P}(x)$, there exists an open set $V \ni x$ such that $\mathcal{P}(V) = \cup_{v \in V} \mathcal{P}(v) \subseteq U$. For the commonly used $\ell_\infty$ perturbations as well as many other commonly used perturbation sets (Yang et al., 2020), these assumptions hold. As an example, in the above notation we would write $\ell_\infty$ perturbations as $\mathcal{P}(x) = \{x' \in B_1 : \|x' - x\|_\infty \leq \tau\}$.

Let $f : B_1 \to \mathbb{R}$ be a predictor. We will let $\ell_c$ be any nonincreasing convex loss with continuous derivative. The adversarial loss is $\ell_A(x, y, f) = \max_{x' \in \mathcal{P}(x)} \ell_c(yf(x'))$, and the adversarial convex risk is $\mathcal{R}_A(f) = \int \ell_A(x, y, f) \, d\mu(x, y)$. For convenience, define $f^+(x) = \sup_{x' \in \mathcal{P}(x)} f(x')$ and $f^-(x) = \inf_{x' \in \mathcal{P}(x)} f(x')$, the worst-case values for perturbations in $\mathcal{P}(x)$ when $y = -1$ and $y = 1$, respectively. Then we can write the adversarial zero-one risk as

$$\mathcal{R}_{\mathrm{AZ}}(f) := \int \left( \mathbb{1}[y = +1]\mathbb{1}[f^-(x) < 0] + \mathbb{1}[y = -1]\mathbb{1}[f^+(x) \geq 0] \right) d\mu(x, y).$$

To relate the adversarial convex and zero-one risks, we will use reweighted adversarial zero-one risks $\mathcal{R}_{\mathrm{AZ}}^t(f)$ as an intermediate quantity, defined as follows. The adversarial zero-one risk when the $+1$ labels have weight $(-\ell_c'(t))$ and the $-1$ labels have weight $(-\ell_c'(-t))$ is

$$\mathcal{R}_{\mathrm{AZ}}^t(f) := \int \left( \mathbb{1}[y = +1]\mathbb{1}[f^-(x) < 0](-\ell_c'(t)) \right.$$
$$\left. + \mathbb{1}[y = -1]\mathbb{1}[f^+(x) \geq 0](-\ell_c'(-t)) \right) d\mu(x, y).$$

### 3.2 RESULTS

We present a number of structural properties of optimal adversarial predictors. The key insight will be to write the global adversarial convex loss in terms of global adversarial zero-one losses, as follows.

**Lemma 3.1.** *For any predictor $f$, $\mathcal{R}_A(f) = \int_{-\infty}^{\infty} \mathcal{R}_{AZ}^t(f - t)\,dt$.*

$\mathcal{R}_{AZ}^t(f - t)$ is an intuitive quantity to consider for the following reason. In the non-adversarial setting, a predictor outputting a value of $f(x)$ corresponds to a prediction of $\frac{(-\ell_c'(-f(x)))}{(-\ell_c'(f(x)))+(-\ell_c'(-f(x)))}$ of the labels being $+1$ at that point. If $+1$ labels are given weight $(-\ell_c'(t))$ and $-1$ labels weight $(-\ell_c'(-t))$, then $f(x)$ would predict at least half the labels being $+1$ if and only if $\frac{(-\ell_c'(-f(x)))}{(-\ell_c'(f(x)))+(-\ell_c'(-f(x)))} \geq \frac{(-\ell_c'(-t))}{(-\ell_c'(t))+(-\ell_c'(-t))}$. As a result, the optimal non-adversarial predictor is also an optimal non-adversarial zero-one classifier at thresholds $t$ for the corresponding reweighting of $+1$ and $-1$ labels. Even though we won't be able to rely on the same local analysis as our intuition above, it turns out the same thing is globally true in the adversarial setting.

**Lemma 3.2.** *There exists a predictor $g : X \to \mathbb{R}$ such that $\mathcal{R}_A(g)$ is minimal. For any such predictor, $\mathcal{R}_{AZ}^t(g - t) = \inf_f \mathcal{R}_{AZ}^t(f)$ for all $t \in \mathbb{R}$.*

Note that the predictor in Lemma 3.2 is not necessarily unique. For instance, the predictor's value at a particular point $x$ might not matter to the adversarial risk because there are no points in the underlying distribution whose perturbation sets include $x$. To prove Lemma 3.2, we will use optimal adversarial zero-one predictors to construct an optimal adversarial convex loss predictor $g$. Conversely, we can also use an optimal adversarial convex loss predictor to construct optimal adversarial zero-one predictors. In general, the predictors we find will not exactly have the optimal adversarial convex loss. For these predictors we have the following error gap.

**Theorem 3.3.** *Suppose there exists $s \geq 1$ and $c > 0$ such that*

$$G_{\ell_c}(p) := \ell_c(0) - \inf_{z \in \mathbb{R}} \left( p\ell_c(z) + (1-p)\ell_c(-z) \right) \geq \left( \frac{|2p-1|}{c} \right)^s.$$

*Then for any predictor $g$,*

$$\inf_t \mathcal{R}_{AZ}(g - t) - \inf_{h\ meas.} \mathcal{R}_{AZ}(h) \leq \frac{3^{1-\frac{1}{s}}c}{2} \left( \mathcal{R}_A(g) - \inf_{h\ meas.} \mathcal{R}_A(h) \right)^{1/s}.$$

The idea of using $G_{\ell_c}(p)$ comes from Zhang (2004). When we explicitly set $\ell_c$ to be the logistic loss in Section 4, we can choose $s = 2$ and $c = \sqrt{2}$ (Zhang, 2004). While similar bounds exist in the non-adversarial case with $\mathcal{R}_{AZ}(g)$ instead of $\inf_t \mathcal{R}_{AZ}(g - t)$ (Zhang, 2004; Bartlett et al., 2006), the analogue with $\mathcal{R}_{AZ}(g - t) - \inf_{h\ meas.} \mathcal{R}_{AZ}(h)$ appearing on the left-hand side is false in the adversarial setting, which can be seen as follows. Consider a uniform distribution of $(x, y)$ pairs over $\{(\pm1, \pm1)\}$, and suppose $\{-1, +1\} \in \mathcal{P}(-1) \cap \mathcal{P}(+1)$. Then the optimal adversarial convex risk is $\ell_c(0)$ given by $f(x) = 0$, and the optimal adversarial zero-one risk is $1/2$ given by $\text{sgn}(f(x)) = +1$. However, for $\epsilon > 0$ the predictor $g_\epsilon$ with $g_\epsilon(+1) = \epsilon$, $g_\epsilon(-1) = -\epsilon$, and $g_\epsilon(x) \in [-\epsilon, \epsilon]$ everywhere else gives adversarial convex risk $\frac{1}{2}\left(\ell_c(\epsilon) + \ell_c(-\epsilon)\right)$ and adversarial zero-one risk 1. This results in

$$\mathcal{R}_{AZ}(g_\epsilon) - \inf_{h\ meas.} \mathcal{R}_{AZ}(h) \xrightarrow[\epsilon \to 0]{} 1/2,$$
$$\mathcal{R}_A(g_\epsilon) - \inf_{h\ meas.} \mathcal{R}_A(h) \xrightarrow[\epsilon \to 0]{} 0,$$

demonstrating the necessity for some change compared to the analogous non-adversarial bound. As this example shows, getting arbitrarily close to the optimal adversarial convex risk does not guarantee getting arbitrarily close to the optimal adversarial zero-one risk. This inadequacy of convex losses in the adversarial setting has been noted in prior work (Bao et al., 2020; Awasthi et al., 2021a;c; 2022a;b), leading them to suggest the use of non-convex losses. However, as Theorem 3.3 shows, we can circumvent this inadequacy if we allow the choice of an optimal (possibly nonzero) threshold.

While we have compared against optimal measurable predictors here, in Section 4 we will use continuous predictors. This presents a potential problem, as there may be a gap between the adversarial risks achievable by measurable and continuous predictors. It turns out this is not the case, as the following lemma shows.

**Lemma 3.4.** *For the adversarial risk, comparing against all continuous functions is equivalent to comparing against all measurable functions. That is, $\inf_{g\ cts.} \mathcal{R}_A(g) = \inf_{h\ meas.} \mathcal{R}_A(h)$.*

In the next section, we will use Lemma 3.4 to compare trained continuous predictors against all measurable functions.

# 4 ADVERSARIAL TRAINING

Theorem 3.3 shows that with optimally chosen thresholds, to achieve nearly optimal adversarial zero-one risk it suffices to achieve nearly optimal adversarial convex risk. However, it is unclear how to find such a predictor. In this section we remedy this issue, proving bounds on the adversarial convex risk when adversarial training is used on shallow ReLU networks. In particular, we show with appropriately chosen parameters we can achieve adversarial convex risk that is arbitrarily close to optimal. Unlike Section 3, our results here will be specific to the logistic loss.

## 4.1 SETTING

Training points $(x_k, y_k)_{k=1}^n$ are drawn from the distribution $\mathcal{D}$. Note that $\|x_k\| \leq 1$, where by default we use $\|\cdot\|$ to denote the $\ell_2$ norm. We will let $\tau = \sup\{\|x' - x\|_2 : x' \in \mathcal{P}(x), x \in B_1\}$ be the maximum $\ell_2$ norm of the adversarial perturbations. By our restrictions on the perturbation sets, we have $0 \leq \tau \leq 2$. Throughout this section we will set $\ell_c$ to be the logistic loss $\ell(z) = \ln(1 + e^{-z})$. The empirical adversarial loss and risk are $\ell_{A,k}(f) = \ell_A(x_k, y_k, f)$ and $\widehat{\mathcal{R}}_A(f) = \frac{1}{n}\sum_{k=1}^n \ell_{A,k}(f)$. The predictors will be shallow ReLU networks of the form $f(W; x) = \frac{\rho}{\sqrt{m}}\sum_{j=1}^m a_i\sigma(w_j^\intercal x)$, where $W$ is an $m \times d$ matrix, $w_j^\intercal$ is the $j$th row of $W$, $\sigma(z) = \max(0, z)$ is the ReLU, $a_i \in \{\pm 1\}$ are initialized uniformly at random, and $\rho$ is a temperature parameter that we can set. Out of all of these parameters, only $W$ will be trained. The initial parameters $W_0$ will have entries initialized from standard Gaussians with variance 1, which we then train to get future iterates $W_i$. We will frequently use the features of $W_i$ for other parameters $W$; that is, we will consider $f^{(i)}(W; x) = \langle \nabla f(W_i; x), W \rangle$, where the gradient is taken with respect to the matrix, not the input. Note that $f$ is not differentiable at all points. When this is the case by $\nabla f$ we mean some choice of $\nabla f \in \partial f$, the Clarke differential. For notational convenience we define $\widehat{\mathcal{R}}_A(W) = \widehat{\mathcal{R}}_A(f(W; \cdot))$, $\mathcal{R}_A(W) = \mathcal{R}_A(f(W; \cdot))$, $\widehat{\mathcal{R}}_A^{(i)}(W) = \widehat{\mathcal{R}}_A(f^{(i)}(W; \cdot))$, and $\mathcal{R}_A^{(i)}(W) = \mathcal{R}_A(f^{(i)}(W; \cdot))$. Our adversarial training will be as follows. To get the next iterate $W_{i+1}$ from $W_i$ for $i \geq 0$ we will use gradient descent with $W_{i+1} = W_i - \eta\nabla\widehat{\mathcal{R}}_A(W_i)$. Normally adversarial training occurs in two steps:

1. For each $k$, find $x_k' \in \mathcal{P}(x)$ such that $\ell(y_k f(W_i; x_k'))$ is maximized.
2. Perform a gradient descent update using the adversarial inputs found in the previous step: $W_{i+1} = W_i - \eta\nabla\left(\frac{1}{n}\sum_{k=1}^n \ell(x_k', y_k, f(W_i; \cdot))\right)$.

Step 1 is an adversarial attack, in practice done with a method such as PGD that does not necessarily find an optimal attack. However, we will assume the idealized scenario where we are able to find an optimal attack. Our goal will be to find a network that has low risk with respect to optimal adversarial attacks. That is, we want to find $f$ such that $\mathcal{R}_A(f)$ is as small as possible.

## 4.2 RESULTS

Our adversarial training theorem will compare the risk we obtain to that of arbitrary reference parameters $Z \in \mathbb{R}^{m \times d}$, which we will choose appropriately when we apply this theorem in Corollaries 4.2 and 4.3. To get near-optimal risk, we will apply our early stopping criterion of running gradient descent until $\|W_i - W_0\| > 2R_Z$, where $R_Z \geq \max\{1, \eta\rho, \|Z - W_0\|\}$, a quantity we assume knowledge of, at which point we will stop. It is possible this may never occur — in that case, we will stop at some time step $t$, which is a parameter we are free to choose. We just need to choose $t$ sufficiently large to allow for enough training to occur. The iterate we choose as our final model will be the one with the best training risk.

**Theorem 4.1.** *Let $m \geq \ln(emd)$ and $\eta\rho^2 < 2$. For any $Z \in \mathbb{R}^{m \times d}$, let $R_Z \geq \max\{1, \eta\rho, \|Z - W_0\|\}$ and $W_{\leq t} = \arg\min\{\widehat{\mathcal{R}}_A(W_i) : 0 \leq i \leq t, \|W_j - W_0\| \leq 2R_Z \quad \forall j \leq i\}$. Then with probability at least $1 - 12\delta$,*

$$\mathcal{R}_A(W_{\leq t}) \leq \frac{2}{2 - \eta\rho^2}\mathcal{R}_A^{(0)}(Z) + \widetilde{\mathcal{O}}\left(\left(\frac{1}{2 - \eta\rho^2}\right)\left(\frac{R_Z^2}{\eta t} + \frac{\rho R_Z\left(d + \sqrt{\tau m}\right)}{\sqrt{n}} + \frac{\rho R_Z^{4/3} d^{1/3}}{m^{1/6}}\right)\right),$$

*where $\widetilde{\mathcal{O}}$ suppresses $\ln(n), \ln(m), \ln(d), \ln(1/\delta), \ln(1/\tau)$ terms.*

In Corollary 4.2 we will show we can set parameters so that all error terms are arbitrarily small.

The early stopping criterion $\|W_i - W_0\| > 2R_Z$ will allow us to get a good generalization bound, as we will show that all iterates then have $\|W_i - W_0\| \leq 2R_Z + \eta\rho$. When the early stopping criterion is met, we will be able to get a good optimization bound. When it is not, choosing $t$ large enough allows us to do so.

It may be concerning that we require knowledge of $R_Z$, as otherwise the algorithm changes depending on which reference parameters we use. However, in practice we could instead use a validation set and instead of choosing the model with the best training risk, we could choose the model with the best validation risk, which would remove the need for knowing $R_Z$. Ultimately, our assumption of knowing $R_Z$ is there to simplify the analysis and highlight other aspects of the problem, and we leave dropping this assumption to future work.

To compare against all continuous functions, we will use the universal approximation of infinite-width neural networks (Barron, 1993). We use a specific form that adapts Barron's arguments to give an estimate of the complexity of the infinite-width neural network (Ji et al., 2019). We consider infinite-width networks of the form $f(x; U_\infty) := \int \langle U_\infty(v), x \mathbb{1}[v^\top x \geq 0] \rangle \, \mathrm{d}\mathcal{N}(v)$, where $U_\infty : \mathbb{R}^d \to \mathbb{R}^d$ parameterizes the network, and $\mathcal{N}$ is a standard $d$-dimensional Gaussian distribution. We will choose an infinite-width network $f(\cdot; U_\infty^\epsilon)$ with a finite complexity measure $\sup_x \|U_\infty^\epsilon(x)\|$ that is $\epsilon$-close to a near-optimal continuous function. Letting $R_\epsilon := \max\{\rho, \eta\rho^2, \sup_x \|U_\infty^\epsilon(x)\|\}$, with high probability we can extract a finite-width network $Z$ close to the infinite-width network whose distance from $W_0$ is at most $R_Z = R_\epsilon/\rho$. Note that our assumed knowledge of $R_Z$ is equivalent to assuming knowledge of $R_\epsilon$. From Lemma 3.4 we know continuous functions can get arbitrarily close to the optimal adversarial risk over all measurable functions, so we can instead compare against all measurable functions.

We immediately have an issue with our formulation — the comparator in Theorem 4.1 is homogeneous. To have any hope of predicting general functions, we need biases. We simulate biases by adding a dummy dimension to the input, and then normalizing. That is, we transform the input $x \to \frac{1}{\sqrt{2}}(x; 1)$. The dummy dimension, while part of the input to the network, is not truly a part of the input, and so we do not allow adversarial perturbations to affect this dummy dimension.

**Corollary 4.2.** *Let $\epsilon > 0$. Then there exists a finite $R_\epsilon \geq \max\{\rho, \eta\rho^2\}$ representing the complexity measure of an infinite-width network that is within $\epsilon$ of the optimal adversarial risk. Then with probability at least $1 - \delta$, setting*

$$\rho = \widetilde{\Theta}(\epsilon), \qquad \eta = \widetilde{\Theta}(1/\epsilon), \qquad t = \widetilde{\Omega}\left(\frac{R_\epsilon^2}{\epsilon^2}\right), \qquad m = \widetilde{\Omega}\left(\frac{R_\epsilon^8}{\epsilon^6\rho^2}\right),$$

*with $n$ satisfying*

$$n = \widetilde{\Omega}\left(\max(1, \tau m)R_\epsilon^2/\epsilon^2\right),$$

*where $\widetilde{\Theta}, \widetilde{\Omega}$ suppresses $\ln(R_\epsilon), \ln(1/\epsilon), \ln(1/\delta)$ terms, we have*

$$\mathcal{R}_A(W_{\leq t}) \leq \inf\{\mathcal{R}_A(g) : g \text{ measurable}\} + \mathcal{O}(\epsilon).$$

Once again, it may be concerning that we require knowledge of the complexity of the data distribution for Corollary 4.2. However, the following result demonstrates that we are effectively guaranteed to converge to the optimal risk as $n \to \infty$, as long as parameters are set appropriately.

**Corollary 4.3.** *If we set*

$$\rho^{(n)} = n^{-1/6}, \qquad \eta^{(n)} = n^{1/6}, \qquad t^{(n)} = n, \qquad m^{(n)} = n^{1/2},$$

*then*

$$\mathcal{R}_A(W_{\leq t}^{(n)}) \xrightarrow{n \to \infty} \inf\{\mathcal{R}_A(g) : g \text{ measurable}\}$$

*almost surely.*

## 4.3 PROOF SKETCH OF THEOREM 4.1

The proof has two main components: a generalization bound and an optimization bound. We describe both in further detail below.

### 4.3.1 GENERALIZATION

Let $\tilde{\tau} := \sqrt{2\tau} + \left(\frac{32\tau \ln(n/\delta)}{m}\right)^{1/4} + \tau\sqrt{2}$. We prove a new near-initialization generalization bound.

**Lemma 4.4.** *If $B \geq 1$ and $m \geq \ln(emd)$, then with probability at least $1 - 5\delta$,*

$$\sup_{\|V - W_0\| \leq B} \left|\mathcal{R}_A^{(0)}(V) - \widehat{\mathcal{R}}_A^{(0)}(V)\right| \leq 2\frac{\rho B}{\sqrt{n}} + 2\frac{\rho B\tilde{\tau}}{\sqrt{n}}\left(1 + \sqrt{m\ln\left(\frac{mn}{\tilde{\tau}^2}\right)}\right)$$

$$+ \frac{77\rho Bd\ln^{3/2}(4em^2d^3/\delta)}{\sqrt{n}}.$$

The key term to focus on is the middle term. Note that $\tilde{\tau}$ is a quantity that grows with the perturbation radius $\tau$, and importantly is 0 when $\tau$ is 0. When we are in the non-adversarial setting ($\tau = 0$), the middle term is dropped and we recover a Rademacher bound qualitatively similar to Lemma A.8 in (Ji et al., 2021). In the general setting when $\tau$ is a constant, so is $\tilde{\tau}$, resulting in an additional dependence on the width of the network.

Lemma 4.4 will easily follow from the following Rademacher complexity bound.

**Lemma 4.5.** *Define $\mathcal{V} = \{V : \|V - W_0\| \leq B\}$, and let*

$$\mathcal{F} = \{x_k \mapsto \min_{x_k' \in \mathcal{P}(x_k)} y_k \left\langle \nabla f(x_k'; W_0), V \right\rangle : V \in \mathcal{V}\}.$$

*Then with probability at least $1 - 3\delta$,*

$$\mathrm{Rad}(\mathcal{F}) \leq \frac{\rho B}{\sqrt{n}} + \frac{\rho B\tilde{\tau}\left(1 + \sqrt{m\ln\left(\frac{mn}{\tilde{\tau}^2}\right)}\right)}{\sqrt{n}}.$$

In the setting of linear predictors with normed ball perturbations, an exact characterization of the perturbations can be obtained, leading to some of the Rademacher bounds in Yin et al. (2018); Khim & Loh (2019); Awasthi et al. (2020). In our setting, it is unclear how to get a similar exact characterization. Instead, we prove Lemma 4.5 by decomposing the adversarially perturbed network into its nonperturbed term and the value added by the perturbation. The nonperturbed term can then be handled by a prior Rademacher bound for standard networks. The difficult part is in bounding the complexity added by the perturbation. A naive argument would worst-case the adversarial perturbations, resulting in a perturbation term that scales linearly with $m$ and does not decrease with $n$. Obtaining the Rademacher bound that appears here requires a better understanding of the adversarial perturbations.

In comparison to simple linear models, we use a more sophisticated approach, utilizing our understanding of linearized models. Because we are using the features of the initial network, for a particular perturbation at a particular point, the same features are used across all networks. As the parameter distance between all networks is close, the same perturbation achieves similar effects. We also control the change in features caused by the perturbations, which is given by Lemma A.9. Having bounded the effect of the perturbation term, we then apply a covering argument over the parameter space to get the Rademacher bound.

### 4.3.2 OPTIMIZATION

Our optimization bound is as follows.

**Lemma 4.6.** *Let $R_{gd} := 2R_Z + \eta\rho$, with $\eta\rho^2 < 2$. Then with probability at least $1 - \delta$,*

$$\widehat{\mathcal{R}}_A(W_{\leq t}) \leq \frac{2}{2 - \eta\rho^2}\widehat{\mathcal{R}}_A^{(0)}(Z) + \frac{1}{t}\left[\frac{R_Z^2}{2\eta - \eta^2\rho^2}\right]$$

$$+ \frac{1}{m^{1/6}}\left(\frac{2}{2 - \eta\rho^2}\right)\left[52\rho R_{gd}^{4/3}d^{1/4}\ln(ed^2m^2/\delta)^{1/4} + \frac{178\rho R_{gd}d^{1/3}\ln(ed^3m^2/\delta)^{1/3}}{m^{1/12}}\right.$$

$$\left. + \frac{5\rho\sqrt{\ln(1/\delta)}}{dm^{5/6}}\right].$$

The main difference between the adversarial case here and the non-adversarial case in Ji et al. (2021) is in appropriately bounding $\|\nabla \widehat{\mathcal{R}}_A(W)\|$, which is Lemma A.10. In order to do so, we utilize a relation between adversarial and non-adversarial losses. The rest of the adversarial optimization proofs follow similarly to the non-adversarial case, although simplified because we assume that $R_Z$ is known.

## 5 DISCUSSION AND OPEN PROBLEMS

This paper leaves open many potential avenues for future work, several of which we highlight below.

**Early stopping.** Early stopping played a key technical role in our proofs, allowing us to take advantage of properties that hold in a near-initialization regime, as well as achieve a good generalization bound. However, is early stopping necessary?

The necessity of early stopping is suggested by the phenomenon of robust overfitting (Rice et al., 2020), in which the adversarial training error continues to decrease, but the adversarial test error dramatically increases after a certain point. Early stopping is one method that allows adversarial training to avoid this phase, and achieve better adversarial test error as a result. However, it should be noted that early stopping is necessary in this work to stay in the near-initialization regime, which likely occurs much earlier than the robust overfitting phase.

**Underparameterization.** Our generalization bound increases with the width. As a result, to get our generalization bound to converge to 0 we required the width to be sublinear in the number of training points. Is it possible to remove this dependence on width?

Recent works suggest that some sort of dependence on the width may be necessary. In the setting of linear regression, overparameterization has been shown to hurt adversarial generalization for specific types of networks (Hassani & Javanmard, 2022; Javanmard et al., 2020; Donhauser et al., 2021). Note that in this simple data setting a simple network can fit the data, so underparameterization does not hurt the approximation capabilities of the network.

However, in a more complicated data setting, Madry et al. (2019) notes that increased width helps with the adversarial test error. One explanation is that networks need to be sufficiently large to approximate an optimal robust predictor, which may be more complicated than optimal nonrobust predictors. Indeed, they note that smaller networks, under adversarial training, would converge to the trivial classifier of predicting a single class. Interestingly, they also note that width helps more when the adversarial perturbation radius is small. This observation is reflected in our generalization bound, since the dependence on width is tied to the perturbation term. If the perturbation radius is small, then a large width is less harmful to our generalization bound. We propose further empirical exploration into how the width affects generalization and approximation error, and how other factors influence this relationship. This includes investigating whether a larger perturbation radius causes larger widths to be more harmful to the generalization bound, the effect of early stopping on these relationships, and how the approximation error for a given width changes with the perturbation radius.

**Using weaker attacks.** Within our proof we used the assumption that we had access to an optimal adversarial attack. In turn, we got a guarantee against optimal adversarial attacks. However, in practice we do not know of a computationally efficient algorithm for generating optimal attacks. Could we prove a similar theorem, getting a guarantee against optimal attacks, using a weaker attack like PGD in our training algorithm? If this was the case, then we would be using a weaker attack to successfully defend against a stronger attack. Perhaps this is too much to ask for — could we get a guarantee against PGD attacks instead?

**Transferability to other settings.** We have only considered the binary classification setting here. A natural extension would be to consider the same questions in the multiclass setting, where there are three (or more) possible labels. In addition, our results in Section 4 only hold for shallow ReLU networks in a near-initialization regime. To what extent do the relationships observed here transfer to other settings, such as training state-of-the-art networks? For instance, does excessive overparameterization beyond the need to capture the complexity of the data hurt adversarial robustness in practice?

ACKNOWLEDGMENTS

The authors are grateful for support from the NSF under grant IIS-1750051.

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

## A    APPENDIX

### A.1    OPTIMAL ADVERSARIAL PREDICTOR AND APPROXIMATION PROOFS

In this section we prove various properties of optimal adversarial predictors. First we need to handle some basic limits.

Recall the definition of $\mathcal{R}_{\text{AZ}}^t(f)$:

$$\mathcal{R}_{\text{AZ}}^t(f) := \int \left( \mathbb{1}[y = +1]\mathbb{1}[f^-(x) < 0](-\ell_c'(t)) + \mathbb{1}[y = -1]\mathbb{1}[f^+(x) \geq 0](-\ell_c'(-t)) \right) \mathrm{d}\mu(x, y).$$

The following lemmas are various forms of continuity for $\mathcal{R}_{\text{AZ}}^t(f)$, making use of the continuity of $\ell_c'$.

**Lemma A.1.** *For any infinite family of predictors $g_s : X \to \{-1, +1\}$ indexed by $s \in \mathbb{R}$, and any $t \in \mathbb{R}$,*

$$\lim_{s \to t} |\mathcal{R}_{\text{AZ}}^t(g_s) - \mathcal{R}_{\text{AZ}}^s(g_s)| = 0.$$

*Proof.* By the Dominated Convergence Theorem (Folland, 1999, Theorem 2.24),

$$\lim_{s \to t} |\mathcal{R}_{\text{AZ}}^t(g_s) - \mathcal{R}_{\text{AZ}}^s(g_s)|$$

$$= \lim_{s \to t} \left| \int \mathbb{1}[y = +1]\mathbb{1}[g_s^-(x) < 0] \left( (-\ell_c'(t)) - (-\ell_c'(s)) \right) \right.$$

$$\left. + \mathbb{1}[y = -1]\mathbb{1}[g_s^+(x) \geq 0] \left( (-\ell_c'(-t)) - (-\ell_c'(-s)) \right) \mathrm{d}\mu(x, y) \right|$$

$$\leq \lim_{s \to t} \int \mathbb{1}[y = +1] \left| (-\ell_c'(t)) - (-\ell_c'(s)) \right|$$

$$+ \mathbb{1}[y = -1] \left| (-\ell_c'(-t)) - (-\ell_c'(-s)) \right| \mathrm{d}\mu(x, y)$$

$$\leq \int \mathbb{1}[y = +1] \lim_{s \to t} \left| (-\ell_c'(t)) - (-\ell_c'(s)) \right|$$

$$+ \mathbb{1}[y = -1] \lim_{s \to t} \left| (-\ell_c'(-t)) - (-\ell_c'(-s)) \right| \mathrm{d}\mu(x, y)$$

$$= 0.$$

$\square$

The following lemma allows us to switch the order of the limit and adversarial risk.

**Lemma A.2.** *For any predictors $g_s : X \to \{-1, +1\}$ indexed by $s \in \mathbb{R}$, and any $t \in \mathbb{R}$,*

$$\lim_{s \to t} \mathcal{R}_{\text{AZ}}^s(g_s) = \mathcal{R}_{\text{AZ}}^t \left( \lim_{s \to t} g_s \right)$$

*when $\lim_{s \to t} g_s$ exists. In particular, when $g_s = f$ for all $s \in \mathbb{R}$, then $\lim_{s \to t} \mathcal{R}_{\text{AZ}}^s(f) = \mathcal{R}_{\text{AZ}}^t(f)$.*

*Proof.* By the Dominated Convergence Theorem (Folland, 1999, Theorem 2.24),

$$\lim_{s \to t} \mathcal{R}_{\text{AZ}}^s(g_s)$$

$$= \lim_{s \to t} \int \mathbb{1}[y = +1]\mathbb{1}[g_s^-(x) < 0](-\ell_c'(s)) + \mathbb{1}[y = -1]\mathbb{1}[g_s^+(x) \geq 0](-\ell_c'(-s)) \, \mathrm{d}\mu(x, y)$$

$$= \int \lim_{s \to t} \left[ \mathbb{1}[y = +1]\mathbb{1}[g_s^-(x) < 0](-\ell_c'(s)) + \mathbb{1}[y = -1]\mathbb{1}[g_s^+(x) \geq 0](-\ell_c'(-s)) \right] \mathrm{d}\mu(x, y)$$

$$= \int \mathbb{1}[y = +1]\mathbb{1}\left[ \lim_{s \to t} g_s^-(x) < 0 \right] (-\ell_c'(t))$$

$$+ \mathbb{1}[y = -1]\mathbb{1}\left[ \lim_{s \to t} g_s^+(x) \geq 0 \right] (-\ell_c'(-t)) \, \mathrm{d}\mu(x, y)$$

$$= \mathcal{R}_{\text{AZ}}^t \left( \lim_{s \to t} g_s \right).$$

$\square$

For the rest of this section, let $f_r : X \to \{-1, +1\}$ be optimal adversarial zero-one predictors when the $-1$ labels are given weight $(-\ell'_c(-r))$ and the $+1$ labels are given weight $(-\ell'_c(r))$ ($f_r$ minimizes $\mathcal{R}^r_{\mathrm{AZ}}(f_r)$), for all $r \in \mathbb{R}$. These predictors exist by Theorem 1 of Bhagoji et al. (2019), although they may not be unique. The following lemma states that $\mathcal{R}^r_{\mathrm{AZ}}(f_r)$ is continuous as a function of $r$.

**Lemma A.3.** *For any $t \in \mathbb{R}$, $\lim_{s \to t} \mathcal{R}^s_{\mathrm{AZ}}(f_s) = \mathcal{R}^t_{\mathrm{AZ}}(f_t)$.*

*Proof.* By Lemma A.1,

$$\limsup_{s \to t} \mathcal{R}^s_{\mathrm{AZ}}(f_s) - \mathcal{R}^t_{\mathrm{AZ}}(f_t) = \limsup_{s \to t} \mathcal{R}^s_{\mathrm{AZ}}(f_s) - \mathcal{R}^s_{\mathrm{AZ}}(f_t) \leq 0,$$

$$\liminf_{s \to t} \mathcal{R}^s_{\mathrm{AZ}}(f_s) - \mathcal{R}^t_{\mathrm{AZ}}(f_t) = \liminf_{s \to t} \mathcal{R}^t_{\mathrm{AZ}}(f_s) - \mathcal{R}^t_{\mathrm{AZ}}(f_t) \geq 0.$$

Together, we get $\lim_{s \to t} \mathcal{R}^s_{\mathrm{AZ}}(f_s) = \mathcal{R}^t_{\mathrm{AZ}}(f_t)$. □

The following lemma gives some structure to optimal adversarial zero-one predictors.

**Lemma A.4.** *For any $s \leq t$, $\mathcal{R}^s_{\mathrm{AZ}}(\max(f_t, f_s)) = \mathcal{R}^s_{\mathrm{AZ}}(f_s)$ and $\mathcal{R}^t_{\mathrm{AZ}}(\min(f_t, f_s)) = \mathcal{R}^t_{\mathrm{AZ}}(f_s)$.*

*Proof.* Let $A := \{x : f_s(x) < f_t(x)\}$, and define

$$A^+_s = \{(x, +1) : \mathcal{P}(x) \cap A \neq \emptyset, f_s(\mathcal{P}(x) \setminus A) \subseteq \{+1\}\},$$
$$A^-_s = \{(x, -1) : \mathcal{P}(x) \cap A \neq \emptyset, f_s(\mathcal{P}(x) \setminus A) \subseteq \{-1\}\},$$
$$A^+_t = \{(x, +1) : \mathcal{P}(x) \cap A \neq \emptyset, f_t(\mathcal{P}(x) \setminus A) \subseteq \{+1\}\},$$
$$A^-_t = \{(x, -1) : \mathcal{P}(x) \cap A \neq \emptyset, f_t(\mathcal{P}(x) \setminus A) \subseteq \{-1\}\}.$$

Let $\mu_s(x, y) = \mathbb{1}[y = +1](-\ell'_c(s))\mu(x, +1) + \mathbb{1}[y = -1](-\ell'_c(-s))\mu(x, -1)$ be the associated measures when the $+1$ labels have weight $(-\ell'_c(s))$ and the $-1$ labels have weight $(-\ell'_c(-s))$, and define $\mu_t$ similarly. Then

$$\mu_s(A^+_s) = (-\ell'_c(s))\mu(A^+_s) \geq (-\ell'_c(t))\mu(A^+_s) \geq (-\ell'_c(t))\mu(A^+_t) = \mu_t(A^+_t),$$
$$\mu_s(A^-_s) = (-\ell'_c(-s))\mu(A^-_s) \leq (-\ell'_c(-t))\mu(A^-_s) \leq (-\ell'_c(-t))\mu(A^-_t) = \mu_t(A^-_t).$$

As a result, $\mu_s(A^+_s) - \mu_s(A^-_s) \geq \mu_t(A^+_t) - \mu_t(A^-_t)$.

The reweighted adversarial zero-one loss can be written in terms of the reweighted measures, as follows.

$$\mathcal{R}^t_{\mathrm{AZ}}(f) = \int \mathbb{1}[y = +1]\mathbb{1}[f^-(x) < 0](-\ell'_c(t))$$
$$+ \mathbb{1}[y = -1]\mathbb{1}[f^+(x) \geq 0](-\ell'_c(-t)) \, \mathrm{d}\mu(x, y)$$
$$= \int \mathbb{1}[y = +1]\mathbb{1}[f^-(x) < 0] + \mathbb{1}[y = -1]\mathbb{1}[f^+(x) \geq 0] \, \mathrm{d}\mu_t(x, y).$$

The following lower bound can then be computed.

$$\mathcal{R}^s_{\mathrm{AZ}}(\max(f_t, f_s)) - \mathcal{R}^s_{\mathrm{AZ}}(f_s)$$
$$= \int \mathbb{1}[y = +1] \left( \mathbb{1}[\max(f_t, f_s)^-(x) < 0] - \mathbb{1}[f^-(x) < 0] \right)$$
$$+ \mathbb{1}[y = -1] \left( \mathbb{1}[\max(f_t, f_s)^+(x) \geq 0] - \mathbb{1}[f^+(x) \geq 0] \right) \mathrm{d}\mu_s(x, y)$$
$$= -\mu_s(A^+_s) + \mu_s(A^-_s)$$
$$\geq 0.$$

Similarly,

$$\mathcal{R}^t_{\mathrm{AZ}}(\min(f_t, f_s)) - \mathcal{R}^t_{\mathrm{AZ}}(f_s)$$
$$= \int \mathbb{1}[y = +1] \left( \mathbb{1}[\min(f_t, f_s)^-(x) < 0] - \mathbb{1}[f^-(x) < 0] \right)$$
$$+ \mathbb{1}[y = -1] \left( \mathbb{1}[\min(f_t, f_s)^+(x) \geq 0] - \mathbb{1}[f^+(x) \geq 0] \right) \mathrm{d}\mu_t(x, y)$$
$$= \mu_t(A^+_t) - \mu_t(A^-_t)$$
$$\geq 0.$$

As $0 \geq \mu_s(A_s^+) - \mu_s(A_s^-) \geq \mu_t(A_t^+) - \mu_t(A_t^-) \geq 0$ we must have equality everywhere, giving the desired result. □

Using our understanding of optimal adversarial zero-one predictors, we can construct a predictor that is optimal at all thresholds.

**Lemma A.5.** *There exists $f : X \to \mathbb{R}$ such that $\mathcal{R}_{\mathrm{AZ}}^t(f - t)$ is the minimum possible value for all $t \in \mathbb{R}$.*

*Proof.* Define $f(x) = \sup\{r \in \mathbb{Q} : f_r(x) = +1\}$. To prove $\mathcal{R}_{\mathrm{AZ}}^t(f - t)$ is minimal for all $t \in \mathbb{R}$, we first do so for all $t \in \mathbb{Q}$, and then for all $t \in \mathbb{R} \setminus \mathbb{Q}$.

For any $t \in \mathbb{Q}$, by definition $\mathrm{sgn}(f - t) \geq f_t$. Let $q_1, q_2, \dots$ be an enumeration of $\mathbb{Q} \cap (t, \infty)$. Let $g_0 = f_t$, and recursively define $g_{i+1} = \max(g_i, f_{q_i})$ for all $i \geq 0$. Note that $\lim_{i \to \infty} g_i = \mathrm{sgn}(f - t)$. Inductively applying Lemma A.4 results in $\mathcal{R}_{\mathrm{AZ}}^t(g_i) = \mathcal{R}_{\mathrm{AZ}}^t(f_t)$ for all $i \geq 0$. By the Dominated Convergence Theorem (Folland, 1999, Theorem 2.24),

$$
\begin{aligned}
\mathcal{R}_{\mathrm{AZ}}^t(f - t) &= \mathcal{R}_{\mathrm{AZ}}^t(\mathrm{sgn}(f - t)) \\
&= \mathcal{R}_{\mathrm{AZ}}^t\left(\lim_{i \to \infty} g_i\right) \\
&= \lim_{i \to \infty} \mathcal{R}_{\mathrm{AZ}}^t(g_i) \\
&= \lim_{i \to \infty} \mathcal{R}_{\mathrm{AZ}}^t(f_t) \\
&= \mathcal{R}_{\mathrm{AZ}}^t(f_t).
\end{aligned}
$$

For any $t \in \mathbb{R} \setminus \mathbb{Q}$, note that $\mathrm{sgn}(f - t) = \lim_{\substack{s \uparrow t \\ s \in \mathbb{Q}}} \mathrm{sgn}(f - s)$. By the Dominated Convergence Theorem (Folland, 1999, Theorem 2.24), Lemma A.2, and Lemma A.3,

$$
\begin{aligned}
\mathcal{R}_{\mathrm{AZ}}^t(f - t) &= \mathcal{R}_{\mathrm{AZ}}^t(\mathrm{sgn}(f - t)) \\
&= \mathcal{R}_{\mathrm{AZ}}^t(\lim_{\substack{s \uparrow t \\ s \in \mathbb{Q}}} \mathrm{sgn}(f - s)) \\
&= \lim_{\substack{s \uparrow t \\ s \in \mathbb{Q}}} \mathcal{R}_{\mathrm{AZ}}^s(\mathrm{sgn}(f - s)) \\
&= \lim_{\substack{s \uparrow t \\ s \in \mathbb{Q}}} \mathcal{R}_{\mathrm{AZ}}^s(f_s) \\
&= \mathcal{R}_{\mathrm{AZ}}^t(f_t).
\end{aligned}
$$

□

For any predictor, its adversarial convex loss can be written as a weighted sum of adversarial zero-one losses across thresholds. This then implies the function defined in Lemma A.5 has optimal adversarial convex loss.

*Proof of Lemma 3.1.* Recall the definition of $\mathcal{R}_{\mathrm{A}}(f)$:

$$
\mathcal{R}_{\mathrm{A}}(f) = \int \left[ \mathbb{1}[y = +1]\ell_c(f^-(x)) + \mathbb{1}[y = -1]\ell_c(-f^+(x)) \right] \mathrm{d}\mu(x, y).
$$

Applying the fundamental theorem of calculus and rearranging,

$$\mathcal{R}_{\mathrm{A}}(f) = \int \left[ \mathbb{1}[y = +1] \int_{f^-(x)}^{\infty} (-\ell'_c(t)) \, \mathrm{d}t + \mathbb{1}[y = -1] \int_{-f^+(x)}^{\infty} (-\ell'_c(t)) \, \mathrm{d}t \right] \mathrm{d}\mu(x, y)$$

$$= \int \left[ \mathbb{1}[y = +1] \int_{-\infty}^{\infty} \mathbb{1}[f^-(x) < t](-\ell'_c(t)) \, \mathrm{d}t \right.$$

$$\left. + \mathbb{1}[y = -1] \int_{-\infty}^{\infty} \mathbb{1}[f^+(x) \geq t](-\ell'_c(-t)) \, \mathrm{d}t \right] \mathrm{d}\mu(x, y)$$

$$= \int \left[ \int_{-\infty}^{\infty} \mathbb{1}[y = +1] \mathbb{1}[f^-(x) < t](-\ell'_c(t)) \right.$$

$$\left. + \mathbb{1}[y = -1] \mathbb{1}[f^+(x) \geq t](-\ell'_c(-t)) \, \mathrm{d}t \right] \mathrm{d}\mu(x, y)$$

As the integrand is nonnegative, Tonelli's theorem (Folland, 1999, Theorem 2.37) gives

$$\mathcal{R}_{\mathrm{A}}(f) = \int_{-\infty}^{\infty} \left[ \int \mathbb{1}[y = +1] \mathbb{1}[f^-(x) < t](-\ell'_c(t)) \right.$$

$$\left. + \mathbb{1}[y = -1] \mathbb{1}[f^+(x) \geq t](-\ell'_c(-t)) \, \mathrm{d}\mu(x, y) \right] \mathrm{d}t$$

$$= \int_{-\infty}^{\infty} \mathcal{R}^t_{\mathrm{AZ}}(f - t) \, \mathrm{d}t.$$

$\square$

While optimal adversarial zero-one predictors were used to construct a predictor with optimal adversarial convex loss in Lemma A.5, the reverse is also possible: using a predictor with optimal adversarial convex loss to construct optimal adversarial zero-one predictors.

*Proof of Lemma 3.2.* Let $f$ be the optimal predictor defined in Lemma A.5, which gives existence. Suppose, towards contradiction, that there exists $g : X \to \mathbb{R}$ with minimal $\mathcal{R}_{\mathrm{A}}(g)$ and $t \in \mathbb{R}$ such that $\mathcal{R}^t_{\mathrm{AZ}}(g - t) > \mathcal{R}^t_{\mathrm{AZ}}(f - t)$. By Lemma A.2 $\mathcal{R}^t_{\mathrm{AZ}}(g - t)$ is left continuous as a function of $t$, and by Lemma A.3 $\mathcal{R}^t_{\mathrm{AZ}}(f - t)$ is continuous as a function of $t$. As a result, there exists $\delta, \epsilon > 0$ such that for all $s \in [t - \delta, t]$,

$$\mathcal{R}^s_{\mathrm{AZ}}(g - s) - \mathcal{R}^s_{\mathrm{AZ}}(f - s) \geq \epsilon.$$

Then

$$\mathcal{R}_{\mathrm{A}}(g) - \mathcal{R}_{\mathrm{A}}(f) = \int_{-\infty}^{\infty} \mathcal{R}^s_{\mathrm{AZ}}(g - s) - \mathcal{R}^s_{\mathrm{AZ}}(f - s) \, \mathrm{d}s$$

$$\geq \int_{t-\delta}^{t} \mathcal{R}^s_{\mathrm{AZ}}(g - s) - \mathcal{R}^s_{\mathrm{AZ}}(f - s) \, \mathrm{d}s$$

$$\geq \int_{t-\delta}^{t} \epsilon \, \mathrm{d}s = \delta\epsilon > 0,$$

contradicting the assumption that $\mathcal{R}_{\mathrm{A}}(g)$ was minimal. So we must have $\mathcal{R}^t_{\mathrm{AZ}}(g - t)$ minimal for all $t \in \mathbb{R}$. $\square$

The next two lemmas bound the zero-one loss at different thresholds in terms of the zero-one loss at threshold 0.

**Lemma A.6.** *Let $f$ be the optimal predictor defined in Lemma A.5. Then $\mathcal{R}^t_{\mathrm{AZ}}(f - t) \leq (-\ell'_c(-|t|))\mathcal{R}_{\mathrm{AZ}}(f)$.*

*Proof.*

$$\mathcal{R}_{\mathrm{AZ}}^t(f-t)$$

$$= \int \mathbb{1}[y=+1]\mathbb{1}[f^-(x)<t](-\ell_c'(t)) + \mathbb{1}[y=-1]\mathbb{1}[f^+(x)\geq t](-\ell_c'(-t))\,\mathrm{d}\mu(x,y)$$

$$\leq \int \mathbb{1}[y=+1]\mathbb{1}[f^-(x)<0](-\ell_c'(t)) + \mathbb{1}[y=-1]\mathbb{1}[f^+(x)\geq 0](-\ell_c'(-t))\,\mathrm{d}\mu(x,y)$$

$$\leq \int \mathbb{1}[y=+1]\mathbb{1}[f^-(x)<0](-\ell_c'(-|t|)) + \mathbb{1}[y=-1]\mathbb{1}[f^+(x)\geq 0](-\ell_c'(-|t|))\,\mathrm{d}\mu(x,y)$$

$$= (-\ell_c'(-|t|))\mathcal{R}_{\mathrm{AZ}}(f).$$

$\square$

In contrast to the previous lemma, the following bound works for any predictor.

**Lemma A.7.** *Let $g$ be any predictor. Then $\mathcal{R}_{\mathrm{AZ}}^t(g-t) \geq (-\ell_c'(|t|))\mathcal{R}_{\mathrm{AZ}}(g-t)$.*

*Proof.*

$$\mathcal{R}_{\mathrm{AZ}}^t(g-t)$$

$$= \int \mathbb{1}[y=+1]\mathbb{1}[g^-(x)<t](-\ell_c'(t)) + \mathbb{1}[y=-1]\mathbb{1}[g^+(x)\geq t](-\ell_c'(-t))\,\mathrm{d}\mu(x,y)$$

$$\geq \int \mathbb{1}[y=+1]\mathbb{1}[g^-(x)<t](-\ell_c'(|t|)) + \mathbb{1}[y=-1]\mathbb{1}[g^+(x)\geq t](-\ell_c'(|t|))\,\mathrm{d}\mu(x,y)$$

$$= (-\ell_c'(|t|))\mathcal{R}_{\mathrm{AZ}}(g-t).$$

$\square$

Now we can relate proximity to the optimal adversarial zero-one loss and proximity to the optimal adversarial convex loss.

*Proof of Theorem 3.3.* Let $f$ be the optimal predictor defined in Lemma A.5. Then

$$\mathcal{R}_{\mathrm{A}}(g) - \inf_{h\text{ meas.}} \mathcal{R}_{\mathrm{A}}(h) = \mathcal{R}_{\mathrm{A}}(g) - \mathcal{R}_{\mathrm{A}}(f)$$

$$= \int_{-\infty}^\infty \mathcal{R}_{\mathrm{AZ}}^u(g-u) - \mathcal{R}_{\mathrm{AZ}}^u(f-u)\,\mathrm{d}u$$

$$\geq \int_{-\infty}^\infty \max(0, (-\ell_c'(|u|))\mathcal{R}_{\mathrm{AZ}}(g-u) - (-\ell_c'(-|u|))\mathcal{R}_{\mathrm{AZ}}(f))\,\mathrm{d}u$$

$$\geq \int_{-\infty}^\infty \max(0, (-\ell_c'(|u|))\inf_t \mathcal{R}_{\mathrm{AZ}}(g-t) - (-\ell_c'(-|u|))\mathcal{R}_{\mathrm{AZ}}(f))\,\mathrm{d}u$$

$$= 2\int_0^\infty \max(0, (-\ell_c'(u))\inf_t \mathcal{R}_{\mathrm{AZ}}(g-t) - (-\ell_c'(-u))\mathcal{R}_{\mathrm{AZ}}(f))\,\mathrm{d}u.$$

As $(-\ell_c'(u))\inf_t \mathcal{R}_{\mathrm{AZ}}(g-t) - (-\ell_c'(-u))\mathcal{R}_{\mathrm{AZ}}(f)$ is continuous and nonincreasing as a function of $u$, as well as nonnegative when $u=0$, there exists some $r \in [0,\infty]$ such that

$$\int_0^\infty \max(0, (-\ell_c'(u))\inf_t \mathcal{R}_{\mathrm{AZ}}(g-t) - (-\ell_c'(-u))\mathcal{R}_{\mathrm{AZ}}(f))\,\mathrm{d}u$$

$$= \int_0^r (-\ell_c'(u))\inf_t \mathcal{R}_{\mathrm{AZ}}(g-t) - (-\ell_c'(-u))\mathcal{R}_{\mathrm{AZ}}(f)\,\mathrm{d}u.$$

Letting $p := \frac{\inf_t \mathcal{R}_{AZ}(g-t)}{\mathcal{R}_{AZ}(f) + \inf_t \mathcal{R}_{AZ}(g-t)}$, we find that this occurs when

$$(-\ell_c'(r)) \inf_t \mathcal{R}_{AZ}(g-t) - (-\ell_c'(-r)) \mathcal{R}_{AZ}(f) = 0$$

$$\iff \frac{\ell_c'(r)}{\ell_c'(-r)} = \frac{\mathcal{R}_{AZ}(f)}{\inf_t \mathcal{R}_{AZ}(g-t)}$$

$$\iff \frac{\ell_c'(r) + \ell_c'(-r)}{\ell_c'(-r)} = \frac{\mathcal{R}_{AZ}(f) + \inf_t \mathcal{R}_{AZ}(g-t)}{\inf_t \mathcal{R}_{AZ}(g-t)}$$

$$\iff \frac{\ell_c'(-r)}{\ell_c'(r) + \ell_c'(-r)} = \frac{\inf_t \mathcal{R}_{AZ}(g-t)}{\mathcal{R}_{AZ}(f) + \inf_t \mathcal{R}_{AZ}(g-t)}$$

$$\iff \frac{\ell_c'(-r)}{\ell_c'(r) + \ell_c'(-r)} = p$$

$$\iff \ell_c'(-r) = \left(\ell_c'(r) + \ell_c'(-r)\right) p$$

$$\iff p\ell_c'(r) - (1-p)\ell_c'(-r) = 0.$$

Since $\ell_c$ is convex, this implies $r$ minimizes $p\ell_c(r) + (1-p)\ell_c(-r)$. The integral can then be computed exactly as follows.

$$\mathcal{R}_A(g) - \inf_{h \text{ meas.}} \mathcal{R}_A(h)$$

$$\geq 2 \int_0^r (-\ell_c'(u)) \inf_t \mathcal{R}_{AZ}(g-t) - (-\ell_c'(-u)) \mathcal{R}_{AZ}(f) \, du$$

$$= 2 \left[ \left(\ell_c(0) - \ell_c(r)\right) \inf_t \mathcal{R}_{AZ}(g-t) - \left(\ell_c(-r) - \ell_c(0)\right) \mathcal{R}_{AZ}(f) \right]$$

$$= 2 \left[ \ell_c(0) \left( \mathcal{R}_{AZ}(f) + \inf_t \mathcal{R}_{AZ}(g-t) \right) - \ell_c(r) \inf_t \mathcal{R}_{AZ}(g-t) - \ell_c(-r) \mathcal{R}_{AZ}(f) \right]$$

$$= 2 \left[ \ell_c(0) \left( \mathcal{R}_{AZ}(f) + \inf_t \mathcal{R}_{AZ}(g-t) \right) \right.$$

$$\left. - \left( \ell_c(r) \frac{\inf_t \mathcal{R}_{AZ}(g-t)}{\mathcal{R}_{AZ}(f) + \inf_t \mathcal{R}_{AZ}(g-t)} + \ell_c(-r) \frac{\mathcal{R}_{AZ}(f)}{\mathcal{R}_{AZ}(f) + \inf_t \mathcal{R}_{AZ}(g-t)} \right) \right.$$

$$\left. \left( \mathcal{R}_{AZ}(f) + \inf_t \mathcal{R}_{AZ}(g-t) \right) \right]$$

$$= 2 \left[ \ell_c(0) \left( \mathcal{R}_{AZ}(f) + \inf_t \mathcal{R}_{AZ}(g-t) \right) \right.$$

$$\left. - \left( \ell_c(r) p + \ell_c(-r)(1-p) \right) \left( \mathcal{R}_{AZ}(f) + \inf_t \mathcal{R}_{AZ}(g-t) \right) \right]$$

$$= 2 \left( \mathcal{R}_{AZ}(f) + \inf_t \mathcal{R}_{AZ}(g-t) \right) \left[ \ell_c(0) - \left( p\ell_c(r) + (1-p)\ell_c(-r) \right) \right]$$

$$= 2 \left( \mathcal{R}_{AZ}(f) + \inf_t \mathcal{R}_{AZ}(g-t) \right) G_{\ell_c}(p).$$

Using our assumption of a lower bound on $G_{\ell_c}(p)$ results in

$$\mathcal{R}_A(g) - \inf_{h \text{ meas.}} \mathcal{R}_A(h) \geq 2 \left( \mathcal{R}_{AZ}(f) + \inf_t \mathcal{R}_{AZ}(g-t) \right) \left( \frac{|2p-1|}{c} \right)^s$$

$$= \frac{2}{c^s} \left( \mathcal{R}_{AZ}(f) + \inf_t \mathcal{R}_{AZ}(g-t) \right) \left( \frac{\inf_t \mathcal{R}_{AZ}(g-t) - \mathcal{R}_{AZ}(f)}{\mathcal{R}_{AZ}(f) + \inf_t \mathcal{R}_{AZ}(g-t)} \right)^s$$

$$= \frac{2}{c^s} \frac{\left( \inf_t \mathcal{R}_{AZ}(g-t) - \mathcal{R}_{AZ}(f) \right)^s}{\left( \mathcal{R}_{AZ}(f) + \inf_t \mathcal{R}_{AZ}(g-t) \right)^{s-1}}.$$

Finally, since $\mathcal{R}_{\text{AZ}}(f) \leq 1/2$ and $\inf_t \mathcal{R}_{\text{AZ}}(g - t) \leq 1$,

$$\mathcal{R}_{\text{A}}(g) - \inf_{h \text{ meas.}} \mathcal{R}_{\text{A}}(h) \geq \frac{2^s}{3^{s-1}c^s} \left( \inf_t \mathcal{R}_{\text{AZ}}(g - t) - \mathcal{R}_{\text{AZ}}(f) \right)^s$$

$$= \frac{2^s}{3^{s-1}c^s} \left( \inf_t \mathcal{R}_{\text{AZ}}(g - t) - \inf_{h \text{ meas.}} \mathcal{R}_{\text{AZ}}(h) \right)^s.$$

Rearranging then gives the desired inequality. $\qquad\square$

The following lemmas states that continuous predictors can get arbitrarily close to the optimal adversarial zero-one risk, even if we require the predictors to output the exact label.

**Lemma A.8.** *Define*

$$\mathcal{R}_{\text{EAZ}}(f) = \int \mathbb{1}[yf(\mathcal{P}(x)) \neq \{+1\}] \, \mathrm{d}\mu(x, y),$$

*the adversarial zero-one risk when we require $f$ to exactly output the right label over the entire perturbation set. Then for any measurable $f : X \to \{-1, +1\}$ and any $\epsilon > 0$, there exists continuous $g : X \to [-1, +1]$ such that $\mu(\{(x,y) : yg(\mathcal{P}(x)) \neq \{+1\} \text{ and } yf(\mathcal{P}(x)) = \{+1\}\}) < \epsilon$. In particular, this implies $\inf_{g \text{ cts.}} \mathcal{R}_{\text{EAZ}}(g) = \inf_{h \text{ meas.}} \mathcal{R}_{\text{EAZ}}(h) = \inf_{g \text{ cts.}} \mathcal{R}_{\text{AZ}}(g) = \inf_{h \text{ meas.}} \mathcal{R}_{\text{AZ}}(h).$*

*Proof.* Let

$$A = \{x : f^+(x) = -1\},$$
$$\mathcal{P}(A) = \cup_{x \in A} \mathcal{P}(x),$$
$$B = \{x : f^-(x) = +1\},$$
$$\mathcal{P}(B) = \cup_{x \in B} \mathcal{P}(x).$$

By the inner regularity of $\mu_x$ (Folland, 1999, Theorem 7.8), there exist compact sets $K \subseteq A, L \subseteq B$ such that

$$\mu_x(A) - \mu_x(K) < \epsilon/2,$$
$$\mu_x(B) - \mu_x(L) < \epsilon/2.$$

As $\mathcal{P}$ is upper hemicontinuous and $K$ and $L$ are compact, both $\mathcal{P}(K)$ and $\mathcal{P}(L)$ are also compact (Aliprantis & Border, 2006, Lemma 17.8). Note that they are also disjoint as $\mathcal{P}(K) \cap \mathcal{P}(L) \subseteq \mathcal{P}(A) \cap \mathcal{P}(B) = \emptyset$. By Urysohn's Lemma (Folland, 1999, Lemma 4.32), there exists a continuous function $g : X \to [0, 1]$ such that $g_t(x) = 0$ for all $x \in \mathcal{P}(K_t)$ and $g_t(x) = 1$ for all $x \in \mathcal{P}(L_t)$. The continuous function $2g - 1 : X \to [-1, 1]$ then satisfies

$$\mu(\{(x, y) : y(2g - 1)(\mathcal{P}(x)) \neq \{+1\} \text{ and } yf(\mathcal{P}(x)) = \{+1\}\})$$
$$\leq \big(\mu_x(A \setminus K)\big) + \big(\mu_x(B \setminus L)\big) < \epsilon.$$

Note that we also have

$$\mathcal{R}_{\text{EAZ}}(2g - 1) \leq \mathcal{R}_{\text{EAZ}}(f) + \mu_x(A \setminus K) - \mu_x(B \setminus L) < \mathcal{R}_{\text{EAZ}}(f) + \epsilon.$$

As $f$ and $\epsilon > 0$ were arbitrary, $\inf_{g \text{ cts.}} \mathcal{R}_{\text{EAZ}}(g) \leq \inf_{h \text{ meas.}} \mathcal{R}_{\text{EAZ}}(h)$.

To get the implication, note that $\inf_{g \text{ cts.}} \mathcal{R}_{\text{AZ}}(g) \leq \inf_{g \text{ cts.}} \mathcal{R}_{\text{EAZ}}(g) \leq \inf_{h \text{ meas.}} \mathcal{R}_{\text{EAZ}}(h) = \inf_{h \text{ meas.}} \mathcal{R}_{\text{AZ}}(h) \leq \inf_{g \text{ cts.}} \mathcal{R}_{\text{AZ}}(g)$, so we must have equality everywhere. $\qquad\square$

While the optimal adversarial predictor may be discontinuous, continuous predictors can get arbitrarily close to the optimal adversarial convex risk.

*Proof of Lemma 3.4.* We have $\inf_{g \text{ cts.}} \mathcal{R}_{\text{A}}(g) \geq \inf_{h \text{ meas.}} \mathcal{R}_{\text{A}}(h)$, so it suffices to show $\inf_{g \text{ cts.}} \mathcal{R}_{\text{A}}(g) \leq \inf_{h \text{ meas.}} \mathcal{R}_{\text{A}}(h)$.

Let $f$ be the optimal predictor defined in Lemma A.5. Then $\mathcal{R}_{\text{A}}(f) = \inf_{h \text{ meas.}} \mathcal{R}_{\text{A}}(h)$, so we want to show $\inf_{g \text{ cts.}} \mathcal{R}_{\text{A}}(g) \leq \mathcal{R}_{\text{A}}(f)$.

Let $\epsilon > 0$. Choose $M > 0$ large enough so that $\mathcal{R}_A(\min(\max(f, -M), M)) < \mathcal{R}_A(f) + \epsilon/3$, and let $\bar{f} = \min(\max(f, -M), M)$. As $\ell_c$ is continuous, there exists a finite-sized partition $P = \{p_0, p_1, p_2, \ldots, p_r\}$ with $p_0 = -M$ and $p_r = M$ such that $\ell_c(p_i) - \ell_c(p_{i-1}) \leq \epsilon/3$ and $\ell_c(-p_i) - \ell_c(-p_{i-1}) \leq \epsilon/3$ for all $1 \leq i \leq r$.

By Lemma A.8, for every $p_i$ there exists continuous $g_{p_i} : X \to [-1, +1]$ such that $\mu(\{(x, y) : yg_{p_i}(\mathcal{P}(x)) \neq \{+1\}$ and $y\mathrm{sgn}(f - p_i)(\mathcal{P}(x)) = \{+1\}\}) < \frac{\epsilon}{3r(\ell_c(-M) - \ell_c(M))}$.

Consider the continuous function $g_\epsilon = -M + \sum_{i=1}^r (p_i - p_{i-1}) \frac{g_{p_i} + 1}{2}$, which will be shown to have adversarial risk within $\epsilon$ of the optimal. Define

$$D_i := \{(x, y) : yg_{p_i}(\mathcal{P}(x)) \neq \{+1\} \text{ and } y\mathrm{sgn}(f - p_i)(\mathcal{P}(x)) = \{+1\}\} \qquad \forall i,$$
$$E := \{(x, y) : \ell_A(x, y, g_\epsilon) > \ell_A(x, y, f) + \epsilon/3\}.$$

We will now show that $E \subseteq \cup_{i=1}^r D_i$. Let $(x, y) \notin \cup_{i=1}^r D_i$. Then $yg_{p_i}(\mathcal{P}(x)) \geq y\mathrm{sgn}(\bar{f} - p_i)(\mathcal{P}(x))$ for all $1 \leq i \leq r$. Let $i' = \arg\max_i\{\mathrm{sgn}(\bar{f} - p_i) = +1\}$. Then $yg_\epsilon(\mathcal{P}(x)) \geq \min\{yp_{i'}, yp_{\max\{i'+1, r\}}\}$ and $yf(\mathcal{P}(x)) \leq \max\{yp_{i'}, yp_{\max\{i'+1, r\}}\}$, so

$$\ell_A(x, y, g_\epsilon) \leq \max\{\ell_c(yp_{i'}), \ell_c(yp_{\max\{i'+1, r\}})\} \leq \min\{\ell_c(yp_{i'}), \ell_c(yp_{\max\{i'+1, r\}})\} + \epsilon/3$$
$$\leq \ell_A(x, y, \bar{f}) + \epsilon/3,$$

which implies $(x, y) \notin E$.

Consequently,

$$\mu(E) \leq \sum_{i=1}^r \mu(D_i) < \sum_{i=1}^r \frac{\epsilon}{3r\left(\ell_c(-M) - \ell_c(M)\right)} = \frac{\epsilon}{3\left(\ell_c(-M) - \ell_c(M)\right)}.$$

Combining the bounds results in

$$\inf_{g \text{ cts.}} \mathcal{R}_A(g) \leq \mathcal{R}_A(g_\epsilon)$$
$$\leq \mathcal{R}_A(\bar{f}) + \epsilon/3 + \mu(E)\left(\ell_c(-M) - \ell_c(M)\right)$$
$$< \mathcal{R}_A(f) + \epsilon/3 + \epsilon/3 + \epsilon/3$$
$$= \mathcal{R}_A(f) + \epsilon.$$

As this holds for all $\epsilon > 0$, we have $\inf_{g \text{ cts.}} \mathcal{R}_A(g) \leq \mathcal{R}_A(f)$, completing the proof. $\qquad\square$

## A.2 GENERALIZATION PROOFS

The following lemma controls the difference in features for nearby points, which will be useful when proving our Rademacher complexity bound.

**Lemma A.9.** *With probability at least $1 - 3n\delta$,*

$$\frac{1}{\rho} \max_{\|\delta_k\| \leq \tau} \left\|\nabla f(x_k; W_0) - \nabla f(x_k + \delta_k; W_0)\right\| \leq \sqrt{2\tau} + \left(\frac{32\tau \ln(1/\delta)}{m}\right)^{1/4} + \tau\sqrt{2},$$

*for all $k$.*

*Proof.* As in Lemma A.2 of Ji et al. (2021), with probability at least $1 - 3n\delta$, for any $x_k \neq 0$,

$$\sum_j \mathbb{1}[|w_{0,j}^\intercal x_k| \leq \tau_k \|x_k\|] \leq m\tau_k + \sqrt{8m\tau_k \ln(1/\delta)},$$

and henceforth assume the failure event does not hold. With $\tau_k = \frac{\tau}{\|x_k\|}$ we have, for any $x_k \neq 0$,

$$\sum_j \mathbb{1}[|w_{0,j}^\intercal x_k| \leq \tau] \leq \frac{m\tau}{\|x_k\|} + \sqrt{\frac{8m\tau \ln(1/\delta)}{\|x_k\|}}.$$

As such, define the set $S_k := \left\{ j : \exists \|\delta_k\| \le \tau, \operatorname{sgn}(w_{0,j}^\top x_k) \ne \operatorname{sgn}(w_{0,j}^\top (x_k + \delta_k)) \right\}$, where the preceding concentration inequality implies $|S_k| \le \frac{m\tau}{\|x_k\|} + \sqrt{\frac{8m\tau \ln(1/\delta)}{\|x_k\|}}$ for all $x_k \ne 0$. Then for any $x_k$ (including $x_k = 0$) and any $\|\delta_k\| \le \tau$,

$$
\frac{1}{\rho^2} \left\| \nabla f(x_k; W_0) - \nabla f(x_k + \delta_k; W_0) \right\|^2
$$

$$
= \frac{1}{m} \sum_j \left\| x_k \mathbb{1}[w_{0,j}^\top x_k \ge 0] - (x_k + \delta_k) \mathbb{1}[w_{0,j}^\top (x_k + \delta_k) \ge 0] \right\|^2
$$

$$
\le \frac{2}{m} \sum_j \left\| x_k \mathbb{1}[w_{0,j}^\top x_k \ge 0] - x_k \mathbb{1}[w_{0,j}^\top (x_k + \delta_k) \ge 0] \right\|^2
$$

$$
+ \frac{2}{m} \sum_j \left\| x_k \mathbb{1}[w_{0,j}^\top (x_k + \delta_k) \ge 0] - (x_k + \delta_k) \mathbb{1}[w_{0,j}^\top (x_k + \delta_k) \ge 0] \right\|^2 .
$$

As $S_k$ is exactly the set of indices $j$ over which $\mathbb{1}[w_{0,j}^\top x_k \ge 0]$ could possibly differ from $\mathbb{1}[w_{0,j}^\top (x_k + \delta_k) \ge 0]$, we can restrict the sum in the first term to these indices, resulting in

$$
\frac{1}{\rho^2} \left\| \nabla f(x_k; W_0) - \nabla f(x_k + \delta_k; W_0) \right\|^2
$$

$$
\le \frac{2}{m} \sum_{j \in S_k} \|x_k\|^2 \left( \mathbb{1}[w_{0,j}^\top x_k \ge 0] - \mathbb{1}[w_{0,j}^\top (x_k + \delta_k) \ge 0] \right)^2
$$

$$
+ \frac{2}{m} \sum_j \left\| \delta_k \mathbb{1}[w_{0,j}^\top (x_k + \delta_k) \ge 0] \right\|^2
$$

$$
\le \frac{2|S_k| \|x_k\|^2}{m} + 2\tau^2
$$

$$
\le 2\tau + \sqrt{\frac{32\tau \ln(1/\delta)}{m}} + 2\tau^2,
$$

where we used $\|x_k\| \le 1$ in the last step. Taking the square root of both sides gives us

$$
\frac{1}{\rho} \left\| \nabla f(x_k; W_0) - \nabla f(x_k + \delta_k; W_0) \right\| \le \sqrt{2\tau} + \left( \frac{32\tau \ln(1/\delta)}{m} \right)^{1/4} + \tau\sqrt{2},
$$

completing the proof. □

We will now prove our Rademacher complexity bound.

*Proof of Lemma 4.5.* We have

$$
\begin{aligned}
n\mathrm{Rad}(\mathcal{F}) &= \mathbb{E}_\epsilon \sup_{V \in \mathcal{V}} \sum_{k=1}^n \epsilon_k \min_{x_k' \in \mathcal{P}(x_k)} y_k \left\langle \nabla f(x_k'; W_0), V \right\rangle \\
&= \mathbb{E}_\epsilon \sup_{V \in \mathcal{V}} \sum_{k=1}^n \left( \epsilon_k \left( \min_{x_k' \in \mathcal{P}(x_k)} y_k \left\langle \nabla f(x_k'; W_0), V \right\rangle - y_k \left\langle \nabla f(x_k; W_0), V \right\rangle \right) \right. \\
&\qquad \left. + \epsilon_k y_k \left\langle \nabla f(x_k; W_0), V \right\rangle \right) \\
&\leq \mathbb{E}_\epsilon \sup_{V \in \mathcal{V}} \sum_{k=1}^n \epsilon_k \left( \min_{x_k' \in \mathcal{P}(x_k)} y_k \left\langle \nabla f(x_k'; W_0) - \nabla f(x_k; W_0), V \right\rangle \right) \\
&\qquad + \mathbb{E}_\epsilon \sup_{U \in \mathcal{V}} \sum_{k=1}^n \epsilon_k y_k \left\langle \nabla f(x_k; W_0), U \right\rangle \\
&\leq \mathbb{E}_\epsilon \sup_{V \in \mathcal{V}} \sum_{k=1}^n \epsilon_k \left( \min_{x_k' \in \mathcal{P}(x_k)} y_k \left\langle \nabla f(x_k'; W_0) - \nabla f(x_k; W_0), V \right\rangle \right) \\
&\qquad + \rho B \sqrt{n},
\end{aligned}
$$

where in the last step we use the Rademacher bound provided in the proof of Lemma A.8 from Ji et al. (2021).

We now focus on bounding $\mathbb{E}_\epsilon \sup_{V \in \mathcal{V}} \sum_{k=1}^n \epsilon_k \left( \min_{x_k' \in \mathcal{P}(x_k)} y_k \left\langle \nabla f(x_k'; W_0) - \nabla f(x_k; W_0), V \right\rangle \right)$.

For notational simplicity let $D_k(V) := \min_{x_k' \in \mathcal{P}(x_k)} y_k \left\langle \nabla f(x_k'; W_0) - \nabla f(x_k; W_0), V \right\rangle$, so that the

quantity we want to bound can be rewritten as $\mathbb{E}_\epsilon \sup_{V \in \mathcal{V}} \sum_{k=1}^n \epsilon_k D_k(V)$.

As Rademacher complexity is invariant under constant shifts, we can subtract the constant $C_k$ from $D_k(V)$, where $C_k := \left( \frac{\sup_{V \in \mathcal{V}'} D_k(V) + \inf_{V \in \mathcal{V}'} D_k(V)}{2} \right)$.

With this constant shift, the expression becomes $\mathbb{E}_\epsilon \sup_{V \in \mathcal{V}} \sum_{k=1}^n \epsilon_k \left( D_k(V) - C_k \right) = n\mathrm{Rad}(\mathcal{G})$, where

$\mathcal{G} = \{ x_k \mapsto \min_{x_k' \in \mathcal{P}(x_k)} y_k \left\langle \nabla f(x_k'; W_0) - \nabla f(x_k; W_0), V \right\rangle - C_k : V \in \mathcal{V} \}$.

We will now bound $n\mathrm{Rad}(\mathcal{G})$ using a covering argument in the parameter space $\mathcal{V}$.

Instead of directly finding a covering for the ball of radius $B$, we first find a covering for a cube with side length $2B$ containing the ball. Projecting the cube to the ball then yields a proper covering of the ball, as this mapping is non-expansive. To ensure every point on the surface of the cube is at most $\epsilon$ distance away from a point, we use a grid with scale $2\epsilon/\sqrt{m}$, which results in $\left( \frac{B\sqrt{m}}{\epsilon} \right)^m$ points in the cover. This cover $\mathcal{C}_\epsilon$ has the property that for every $V \in \mathcal{V}$, there is some $U \in \mathcal{C}_\epsilon$ such that $\|V - U\| \leq \epsilon$, since every coordinate of $V$ is $\epsilon/\sqrt{m}$-close to a coordinate in $\mathcal{C}_\epsilon$, and $\|V - U\| = \sqrt{\sum_{i=1}^m (V_i - U_i)^2} \leq \sqrt{\sum_{i=1}^m (\epsilon/\sqrt{m})^2} = \epsilon$. Due to the non-expansive projection mapping from the cube to the sphere, the $\epsilon$-cover for the cube projects to an $\epsilon$-cover for the sphere. As a result, we have an $\epsilon$-cover for the sphere of radius $B$ with $\left( \frac{B\sqrt{m}}{\epsilon} \right)^m$ points.

A geometric $\epsilon$-cover gives only a $\rho\tilde{\tau}\epsilon$-cover in the function space, since for any $V$ and $U$ with $\|V - U\| \le \epsilon$, and any $x_k$,

$$\left\| \left( \min_{x_V \in \mathcal{P}(x_k)} y_k \left\langle \nabla f(x_V; W_0) - \nabla f(x_k; W_0), V \right\rangle - C_k \right) \right.$$

$$\left. - \left( \min_{x_U \in \mathcal{P}(x_k)} y_k \left\langle \nabla f(x_U; W_0) - \nabla f(x_k; W_0), U \right\rangle - C_k \right) \right\|$$

$$\le \sup_{\substack{\|V-U\| \le \epsilon \\ x_U \in \mathcal{P}(x_k)}} \left( \min_{x_V \in \mathcal{P}(x_k)} \left( y_k \left\langle \nabla f(x_V; W_0) - \nabla f(x_k; W_0), V \right\rangle - C_k \right) \right.$$

$$\left. - \left( y_k \left\langle \nabla f(x_U; W_0) - \nabla f(x_k; W_0), U \right\rangle - C_k \right) \right)$$

$$\le \sup_{\substack{\|V-U\| \le \epsilon \\ x_U \in \mathcal{P}(x_k)}} \left( \left( y_k \left\langle \nabla f(x_U; W_0) - \nabla f(x_k; W_0), V \right\rangle - C_k \right) \right.$$

$$\left. - \left( y_k \left\langle \nabla f(x_U; W_0) - \nabla f(x_k; W_0), U \right\rangle - C_k \right) \right)$$

$$= \sup_{\substack{\|V-U\| \le \epsilon \\ x_U \in \mathcal{P}(x_k)}} \left( y_k \left\langle \nabla f(x_U; W_0) - \nabla f(x_k; W_0), V - U \right\rangle \right)$$

$$\le \sup_{x_U \in \mathcal{P}(x_k)} \|\nabla f(x_U; W_0) - \nabla f(x_k; W_0)\| \|V - U\|$$

$$\le \sup_{\|\delta_U\| \le \tau} \|\nabla f(x_U; W_0) - \nabla f(x_k; W_0)\| \|V - U\|$$

$$\le \rho\tilde{\tau}\epsilon,$$

where the last step follows with probability at least $1 - 3\delta$ by Lemma A.9.

As a result, we can get an $\epsilon$-cover in the function space with just $\left( \frac{\rho B \tilde{\tau} \sqrt{m}}{\epsilon} \right)^m$ points.

This bound on the covering number $\mathcal{N}(\mathcal{G}, \epsilon, \|\cdot\|_u) \le \left( \frac{\rho B \tilde{\tau} \sqrt{m}}{\epsilon} \right)^m$ then implies

$$\mathcal{N}(\mathcal{G}, \epsilon, \|\cdot\|_2) \le \mathcal{N}(\mathcal{G}, \epsilon/\sqrt{n}, \|\cdot\|_u) \le \left( \frac{\rho B \tilde{\tau} \sqrt{mn}}{\epsilon} \right)^m,$$

which we can use in a standard parameter-based covering argument (Anthony & Bartlett, 2009) to get

$$n\text{Rad}(\mathcal{G}) \le \inf_{\alpha > 0} \left( \alpha \sqrt{n} + \left( \sup_{U \in \mathcal{G}} \|U\|_2 \right) \sqrt{2 \ln \mathcal{N}(\mathcal{G}, \alpha, \|\cdot\|_2)} \right)$$

$$\le \inf_{\alpha > 0} \left( \alpha \sqrt{n} + \left( \rho B \tilde{\tau} \sqrt{n} \right) \sqrt{2 \ln \left( \frac{\rho B \tilde{\tau} \sqrt{mn}}{\alpha} \right)^m} \right).$$

To calculate $\sup_{U \in \mathcal{G}} \|U\|_2$, note that each of the $n$ entries is bounded above by

$$\sup_{V \in \mathcal{V}'} \left| D_k(V) - C_k \right| = \max \left\{ \sup_{V \in \mathcal{V}'} D_k(V) - C_k, - \left( \inf_{U \in \mathcal{V}'} D_k(U) - C_k \right) \right\}$$

$$= \left( \frac{\sup_{V \in \mathcal{V}'} D_k(V) - \inf_{U \in \mathcal{V}'} D_k(U)}{2} \right)$$

$$\le \frac{1}{2} \rho \tilde{\tau} 2B$$

$$= \rho B \tilde{\tau},$$

so $\sup_{U \in \mathcal{G}} \|U\|_2 \leq \rho B \tilde{\tau} \sqrt{n}$.

Setting $\alpha = \rho B \tilde{\tau}$ (let $\alpha \to 0$ if $\tilde{\tau} = 0$) we get

$$n\text{Rad}(\mathcal{G}) \leq \rho B \tilde{\tau} \sqrt{n} + \rho B \tilde{\tau} \sqrt{mn \ln \left( \frac{mn}{\tilde{\tau}^2} \right)}.$$

Putting this together with our previous bound results in

$$n\text{Rad}(\mathcal{F}) \leq \rho B \sqrt{n} + n\text{Rad}(\mathcal{G}) \leq \rho B \sqrt{n} + \rho B \tilde{\tau} \sqrt{n} \left( 1 + \sqrt{m \ln \left( \frac{mn}{\tilde{\tau}^2} \right)} \right),$$

and dividing by $n$ then completes the proof. $\qquad\square$

With our Rademacher complexity bound we can prove our generalization bound, as follows.

*Proof of Lemma 4.4.* By Lemma A.6 part 2 of Ji et al. (2021), with probability at least $1 - \delta$,

$$\sup_{\|V - W_0\| \leq B} \sup_{\|x\| \leq 1} \left| \left\langle \nabla f(x; W_0), V \right\rangle \right| \leq 18 \rho B \ln(emd(1 + 3(md^{3/2})^d)/\delta)$$
$$\leq 18 \rho B d \ln(4em^2 d^3/\delta).$$

By the decreasing monotonicity of the logistic loss,

$$\sup_{\|V - W_0\| \leq B} \sup_{\|x\| \leq 1} \ell_{\text{A}}(x, y, \left\langle \nabla f(\cdot; W_0), V \right\rangle)$$
$$\in [\ell(18 \rho B d \ln(4em^2 d^3/\delta)), \ell(-18 \rho B d \ln(4em^2 d^3/\delta))]$$
$$\subseteq [\ln(2) - 18 \rho B d \ln(4em^2 d^3/\delta), \ln(2) + 18 \rho B d \ln(4em^2 d^3/\delta)].$$

With a bound on the range from above that holds with probability at least $1 - \delta$ and a bound on the Rademacher complexity from Lemma 4.5 that holds with probability at least $1 - 3\delta$, we can now apply a standard Rademacher bound (Shalev-Shwartz & Ben-David, 2014) that holds with probability at least $1 - \delta$ to get that altogether with probability at least $1 - 5\delta$,

$$\sup_{\|V - W_0\| \leq B} \left| \mathcal{R}_{\text{A}}^{(0)}(V) - \widehat{\mathcal{R}}_{\text{A}}^{(0)}(V) \right| \leq 2\text{Rad}(\ell_{\text{A}}(\mathcal{F})) + 3(36 \rho B d \ln(4em^2 d^3/\delta)) \sqrt{\frac{\ln(4/\delta)}{2n}}$$

$$\leq 2\frac{\rho B}{\sqrt{n}} + 2\frac{\rho B \tilde{\tau}}{\sqrt{n}} \left( 1 + \sqrt{m \ln \left( \frac{mn}{\tilde{\tau}^2} \right)} \right) + \frac{77 \rho B d \ln^{3/2}(4em^2 d^3/\delta)}{\sqrt{n}}.$$

$\qquad\square$

### A.3 OPTIMIZATION PROOFS

The following lemma bounds the gradient of the adversarial risk.

**Lemma A.10.** *For any matrix* $W \in \mathbb{R}^{m \times d}$, $\left\| \nabla \widehat{\mathcal{R}}_{\text{A}}(W) \right\| \leq \rho \min \left\{ 1, \widehat{\mathcal{R}}_{\text{A}}(W) \right\}$.

*Proof.* By properties of the logistic loss (Ji & Telgarsky, 2018),

$$
\begin{aligned}
\left\| \nabla \widehat{\mathcal{R}}_{\mathrm{A}}(W) \right\| &= \left\| \frac{1}{n} \sum_{k=1}^{n} \ell' \left( \min_{x_k' \in \mathcal{P}(x_k)} y_k f(W; x_k') \right) \nabla \left( \min_{x_k' \in \mathcal{P}(x_k)} y_k f(W; x_k') \right) \right\| \\
&\leq \frac{1}{n} \sum_{k=1}^{n} \left\| \ell' \left( \min_{x_k' \in \mathcal{P}(x_k)} y_k f(W; x_k') \right) \nabla \left( \min_{x_k' \in \mathcal{P}(x_k)} y_k f(W; x_k') \right) \right\| \\
&= \frac{1}{n} \sum_{k=1}^{n} \left| \ell' \left( \min_{x_k' \in \mathcal{P}(x_k)} y_k f(W; x_k') \right) \right| \left\| \nabla \left( \min_{x_k' \in \mathcal{P}(x_k)} y_k f(W; x_k') \right) \right\| \\
&\leq \frac{1}{n} \sum_{k=1}^{n} \min \left\{ 1, \ell \left( \min_{x_k' \in \mathcal{P}(x_k)} y_k f(W; x_k') \right) \right\} \rho \\
&= \frac{1}{n} \sum_{k=1}^{n} \min \left\{ 1, \ell_{\mathrm{A},k}(f) \right\} \rho \\
&\leq \rho \min \left\{ 1, \widehat{\mathcal{R}}_{\mathrm{A}}(W) \right\}.
\end{aligned}
$$

$\square$

We have the following guarantee when using adversarial training.

**Lemma A.11.** *When adversarially training with step size $\eta$, for any iterate $t$ and any reference parameters $Z \in \mathbb{R}^{m \times d}$,*

$$
\| W_t - Z \|^2 + (2\eta - \eta^2 \rho^2) \sum_{i<t} \widehat{\mathcal{R}}_{\mathrm{A}}(W_i) \leq \| W_0 - Z \|^2 + 2\eta \sum_{i<t} \widehat{\mathcal{R}}_{\mathrm{A}}^{(i)}(Z).
$$

*Proof.* It suffices to show

$$
\| W_{i+1} - Z \|^2 + (2\eta - \eta^2 \rho^2) \widehat{\mathcal{R}}_{\mathrm{A}}(W_i) \leq \| W_i - Z \|^2 + 2\eta \widehat{\mathcal{R}}_{\mathrm{A}}^{(i)}(Z)
$$

for $0 \leq i < t$, as summing the left- and right-hand sides over $0 \leq i < t$ then gives the desired bound. By the definition of $W_{i+1}$,

$$
\| W_{i+1} - Z \|^2 = \| W_i - Z \|^2 - 2\eta \left\langle \nabla \widehat{\mathcal{R}}_{\mathrm{A}}(W_i), W_i - Z \right\rangle + \eta^2 \left\| \nabla \widehat{\mathcal{R}}_{\mathrm{A}}(W_i) \right\|^2.
$$

Note that

$$
\begin{aligned}
-2\eta \left\langle \nabla \widehat{\mathcal{R}}_{\mathrm{A}}(W_i), W_i - Z \right\rangle &= 2\eta \left\langle \nabla \widehat{\mathcal{R}}_{\mathrm{A}}(W_i), Z - W_i \right\rangle \\
&= 2\eta \left\langle \nabla \widehat{\mathcal{R}}_{\mathrm{A}}(W_i), Z - W_i \right\rangle \\
&= 2\eta \left\langle \nabla \widehat{\mathcal{R}}_{\mathrm{A}}^{(i)}(W_i), Z - W_i \right\rangle \\
&\leq 2\eta \left( \widehat{\mathcal{R}}_{\mathrm{A}}^{(i)}(Z) - \widehat{\mathcal{R}}_{\mathrm{A}}^{(i)}(W_i) \right),
\end{aligned}
$$

where the last inequality follows because $\widehat{\mathcal{R}}_{\mathrm{A}}^{(i)}(W) = \widehat{\mathcal{R}}_{\mathrm{A}}(f^{(i)}(W; \cdot))$ is convex in the function space $f^{(i)}(W; \cdot)$, which in turn is linear in $W$, so $\widehat{\mathcal{R}}_{\mathrm{A}}^{(i)}(W)$ is convex in $W$.

Using this bound, in addition to Lemma A.10, gives

$$
\| W_{i+1} - Z \|^2 \leq \| W_i - Z \|^2 + 2\eta \left( \widehat{\mathcal{R}}_{\mathrm{A}}^{(i)}(Z) - \widehat{\mathcal{R}}_{\mathrm{A}}^{(i)}(W_i) \right) + \eta^2 \rho^2 \widehat{\mathcal{R}}_{\mathrm{A}}(W_i),
$$

and rearranging then gives the desired inequality. $\square$

We want to bound $\widehat{\mathcal{R}}_{\mathrm{A}}^{(i)}(Z)$ in terms of $\widehat{\mathcal{R}}_{\mathrm{A}}^{(0)}(Z)$. To do so, we will show that when changing features the value at every point remains close to its original value. Towards this goal, we first show that the features in a small ball do not change much.

**Lemma A.12.** *For any $\|z\| \leq 1$ and any $0 < \epsilon \leq 1/(dm)$, with probability at least $1 - \delta$,*

$$\sup_{\substack{\|x-z\| \leq \epsilon \\ \|x\| \leq 1 \\ \|V-W_0\| \leq R_V}} \|\nabla f(x; V) - \nabla f(z; V)\|$$

$$\leq 7\rho R_V^{1/3} m^{-1/6} \left(\ln(em/\delta)\right)^{1/6} + 12\rho d^{1/6} \epsilon^{1/3} \left(\ln(em/\delta)\right)^{1/3} + 2\rho\epsilon + 15\rho \left(\frac{\ln(edm/\delta)}{m}\right)^{1/4}.$$

*Proof.* For notational convenience let $W := W_0$. First, note that

$$\|\nabla f(x; V) - \nabla f(z; V)\|$$
$$\leq \|\nabla f(x; V) - \nabla f(x; W)\| + \|\nabla f(x; W) - \nabla f(z; W)\| + \|\nabla f(z; W) - \nabla f(z; V)\|.$$

By Lemma A.5 of Ji et al. (2021), with probability at least $1 - \delta$ the middle term is bounded by

$$\|\nabla f(x; W) - \nabla f(z; W)\| \leq 11\rho \left(\frac{\ln(edm/\delta)}{m}\right)^{1/4}.$$

The first and last terms are both bounded by $\sup_{\substack{\|x-z\| \leq \epsilon \\ \|x\| \leq 1 \\ \|V-W\| \leq R_V}} \|\nabla f(x; V) - \nabla f(x; W)\|$, so we will

now focus on bounding this term. Note that

$$\nabla f(x; V) - \nabla f(x; W) = \frac{\rho}{\sqrt{m}} \sum_{j=1}^{m} a_j \left[\mathbb{1}[v_j^\mathsf{T} x \geq 0] - \mathbb{1}[w_j^\mathsf{T} x \geq 0]\right] e_j x^\mathsf{T}.$$

We consider two cases: $\|x\| \leq (k+1)\epsilon$ and $\|x\| > (k+1)\epsilon$, for some $k \geq 1$ to be determined later.

- **Case 1:** $\|x\| \leq (k+1)\epsilon$.

  Then $\|\nabla f(x; V) - \nabla f(x; W)\| \leq \frac{\rho}{\sqrt{m}} \sqrt{m}(k+1)\epsilon = (k+1)\rho\epsilon.$

- **Case 2:** $\|x\| > (k+1)\epsilon$.

  Then $\|z\| > k\epsilon$. With probability at least $1 - m\delta$, $\|w_j\| \leq \sqrt{d} + \sqrt{2\ln(1/\delta)}$ for all $1 \leq j \leq m$. Define

  $$S_1 := \left\{j \in [m] : |w_j^\mathsf{T} z| \leq q\|z\|\right\},$$
  $$S_2 := \left\{j \in [m] : |w_j^\mathsf{T} x| \leq r\|x\|\right\},$$
  $$S_3 := \left\{j \in [m] : \|v_j - w_j\| \geq r\right\},$$
  $$S := S_2 \cup S_3,$$

  where $q := 2\left(r + \frac{1}{k}\sqrt{d} + \frac{1}{k}\sqrt{2\ln(1/\delta)}\right)$, with $r$ a parameter we will choose later. Note that $S_2 \subseteq S_1$, since if $|w_j^\mathsf{T} x| \leq r\|x\|$, then

  $$|w_j^\mathsf{T} z| \leq |w_j^\mathsf{T} x| + \|w_j\|\epsilon \leq r\|x\| + \frac{1}{k}\|w_j\|\|x\| = \left(r + \frac{1}{k}\|w_j\|\right)\|x\|$$
  $$\leq \frac{k+1}{k}\left(r + \frac{1}{k}\|w_j\|\right)\|z\| \leq 2\left(r + \frac{1}{k}\sqrt{d} + \frac{1}{k}\sqrt{2\ln(1/\delta)}\right)\|z\| \leq q\|z\|.$$

By Lemma A.2 part 1 of Ji et al. (2021) we have that with probability at least $1 - 3\delta$, $|S_2| \leq qm + \sqrt{8qm\ln(1/\delta)}$. Within the proof of Lemma A.7 part 1 of Ji et al. (2021) it is shown that

$|S_3| \leq \frac{R_V^2}{r^2}$. So altogether, with probability at least $1 - (m+3)\delta$,

$$
\begin{aligned}
|S| &\leq |S_2| + |S_3| \\
&\leq qm + \sqrt{8qm\ln(1/\delta)} + \frac{R_V^2}{r^2} \\
&\leq 2\left(r + \frac{1}{k}\sqrt{d} + \frac{1}{k}\sqrt{2\ln(1/\delta)}\right)m + \sqrt{16\left(r + \frac{1}{k}\sqrt{d} + \frac{1}{k}\sqrt{2\ln(1/\delta)}\right)m\ln(1/\delta)} \\
&\quad + \frac{R_V^2}{r^2} \\
&\leq 6\left(r + \frac{1}{k}\sqrt{d} + \frac{1}{k}\sqrt{2\ln(1/\delta)}\right)m\sqrt{\ln(e/\delta)} + \frac{R_V^2}{r^2} \\
&\leq 6\left(r + \frac{3}{k}\sqrt{d\ln(e/\delta)}\right)m\sqrt{\ln(e/\delta)} + \frac{R_V^2}{r^2} \\
&= 6rm\sqrt{\ln(e/\delta)} + \frac{18m\sqrt{d}\ln(e/\delta)}{k} + \frac{R_V^2}{r^2}.
\end{aligned}
$$

Setting $r := R_V^{2/3}m^{-1/3}\ln(e/\delta)^{-1/6}$ we get $|S| \leq 7R_V^{2/3}m^{2/3}\left(\ln(e/\delta)\right)^{1/3} + \frac{18m\sqrt{d}\ln(e/\delta)}{k}$.
Substituting this upper bound on $|S|$ results in

$$
\begin{aligned}
\|\nabla f(x;V) - \nabla f(x;W)\| &\leq \frac{\rho}{\sqrt{m}}\sqrt{|S|}\|x\| \leq \frac{\rho}{\sqrt{m}}\sqrt{|S|} \\
&\leq \frac{\rho}{\sqrt{m}}\left(3R_V^{1/3}m^{1/3}\left(\ln(e/\delta)\right)^{1/6} + 5m^{1/2}d^{1/4}k^{-1/2}\sqrt{\ln(e/\delta)}\right) \\
&\leq 3\rho R_V^{1/3}m^{-1/6}\left(\ln(e/\delta)\right)^{1/6} + 5\rho d^{1/4}k^{-1/2}\sqrt{\ln(e/\delta)}.
\end{aligned}
$$

Combining the two cases results in

$$
\begin{aligned}
&\|\nabla f(x;V) - \nabla f(x;W)\| \\
&\leq \max\{(k+1)\rho\epsilon, 3\rho R_V^{1/3}m^{-1/6}\left(\ln(e/\delta)\right)^{1/6} + 5\rho d^{1/4}k^{-1/2}\sqrt{\ln(e/\delta)}\}.
\end{aligned}
$$

After setting $k := d^{1/6}\epsilon^{-2/3}\left(\ln(e/\delta)\right)^{1/3}$ to balance the terms,

$$
\begin{aligned}
&\|\nabla f(x;V) - \nabla f(x;W)\| \\
&\leq \max\{\rho d^{1/6}\epsilon^{1/3}\left(\ln(e/\delta)\right)^{1/3} + \rho\epsilon, 3\rho R_V^{1/3}m^{-1/6}\left(\ln(e/\delta)\right)^{1/6} + 5\rho d^{1/6}\epsilon^{1/3}\left(\ln(e/\delta)\right)^{1/3}\} \\
&\leq 3\rho R_V^{1/3}m^{-1/6}\left(\ln(e/\delta)\right)^{1/6} + 5\rho d^{1/6}\epsilon^{1/3}\left(\ln(e/\delta)\right)^{1/3} + \rho\epsilon.
\end{aligned}
$$

So with probability at least $1 - (m+3)\delta$,

$$
\begin{aligned}
&\sup_{\substack{\|x-z\|\leq\epsilon \\ \|x\|\leq 1 \\ \|V-W\|\leq R_V}} \|\nabla f(x;V) - \nabla f(z;V)\| \\
&\leq 6\rho R_V^{1/3}m^{-1/6}\left(\ln(e/\delta)\right)^{1/6} + 10\rho d^{1/6}\epsilon^{1/3}\left(\ln(e/\delta)\right)^{1/3} + 2\rho\epsilon + 11\rho\left(\frac{\ln(edm/\delta)}{m}\right)^{1/4}.
\end{aligned}
$$

Rescaling the probability of failure, we get that with probability at least $1 - \delta$,

$$
\begin{aligned}
&\sup_{\substack{\|x-z\|\leq\epsilon \\ \|x\|\leq 1 \\ \|V-W\|\leq R_V}} \|\nabla f(x;V) - \nabla f(z;V)\| \\
&\leq 7\rho R_V^{1/3}m^{-1/6}\left(\ln(em/\delta)\right)^{1/6} + 12\rho d^{1/6}\epsilon^{1/3}\left(\ln(em/\delta)\right)^{1/3} + 2\rho\epsilon + 15\rho\left(\frac{\ln(edm/\delta)}{m}\right)^{1/4}.
\end{aligned}
$$

$\square$

Using a sphere covering argument, we bound $\widehat{\mathcal{R}}_{\mathrm{A}}^{(i)}(Z)$ in terms of $\widehat{\mathcal{R}}_{\mathrm{A}}^{(0)}(Z)$, as well as $\mathcal{R}^{(i)}(Z)$ in terms of $\mathcal{R}^{(0)}(Z)$.

**Lemma A.13.** *1. For any $\|z\| \leq 1$ and $R_V \geq 1$ and $R_B \geq 0$, with probability at least $1 - \delta$,*

$$
\sup_{\substack{\|x\| \leq 1 \\ \|V - W_0\| \leq R_V \\ \|B - W_0\| \leq R_B}} \left| \left\langle \nabla f(x; V) - \nabla f(x; W_0), B \right\rangle \right|
$$

$$
\leq \frac{26\rho(R_B + R_V)R_V^{1/3}d^{1/4}\ln(ed^2m^2/\delta)^{1/4}}{m^{1/6}} + \frac{89\rho(R_B + R_V)d^{1/3}\ln(ed^3m^2/\delta)^{1/3}}{m^{1/4}}
$$

$$
+ \frac{5\rho\sqrt{\ln(1/\delta)}}{dm}.
$$

*2. With probability at least $1 - \delta$, simultaneously*

$$
\sup_{\substack{\|W_i - W_0\| \leq R_V \\ \|B - W_0\| \leq R_B}} \left| \widehat{\mathcal{R}}_{\mathrm{A}}^{(i)}(B) - \widehat{\mathcal{R}}_{\mathrm{A}}^{(0)}(B) \right|
$$

$$
\leq \frac{26\rho(R_B + R_V)R_V^{1/3}d^{1/4}\ln(ed^2m^2/\delta)^{1/4}}{m^{1/6}} + \frac{89\rho(R_B + R_V)d^{1/3}\ln(ed^3m^2/\delta)^{1/3}}{m^{1/4}}
$$

$$
+ \frac{5\rho\sqrt{\ln(1/\delta)}}{dm}
$$

*and*

$$
\sup_{\substack{\|W_i - W_0\| \leq R_V \\ \|B - W_0\| \leq R_B}} \left| \mathcal{R}^{(i)}(B) - \mathcal{R}^{(0)}(B) \right|
$$

$$
\leq \frac{26\rho(R_B + R_V)R_V^{1/3}d^{1/4}\ln(ed^2m^2/\delta)^{1/4}}{m^{1/6}} + \frac{89\rho(R_B + R_V)d^{1/3}\ln(ed^3m^2/\delta)^{1/3}}{m^{1/4}}
$$

$$
+ \frac{5\rho\sqrt{\ln(1/\delta)}}{dm}.
$$

*Proof.* 1. For notational convenience let $W := W_0$. For some $0 < \epsilon \leq 1/(dm)$ which we will choose later, instantiate a cover $\mathcal{C}$ at scale $\epsilon/\sqrt{d}$, with $|\mathcal{C}| \leq (\sqrt{d}/\epsilon)^d$. For a given point $x$, let $z \in \mathcal{C}$ denote the closest point in the cover. Then by the triangle inequality,

$$
\sup_{\substack{\|x\| \leq 1 \\ \|V - W\| \leq R_V \\ \|B - W\| \leq R_B}} \left| \left\langle \nabla f(x; V) - \nabla f(x; W), B \right\rangle \right|
$$

$$
\leq \sup_{\substack{\|x\| \leq 1 \\ \|V - W\| \leq R_V \\ \|B - W\| \leq R_B}} \left| \left\langle \nabla f(x; V) - \nabla f(x; W), B \right\rangle - \left\langle \nabla f(z; V) - \nabla f(z; W), B \right\rangle \right|
$$

$$
+ \left| \left\langle \nabla f(z; V) - \nabla f(z; W), B \right\rangle \right|
$$

$$
\leq \sup_{\substack{\|x\| \leq 1 \\ \|V - W\| \leq R_V \\ \|B - W\| \leq R_B}} \left| \left\langle \nabla f(x; V) - \nabla f(z; V), B \right\rangle \right| + \left| \left\langle \nabla f(x; W) - \nabla f(z; W), B \right\rangle \right|
$$

$$
+ \left| \left\langle \nabla f(z; V) - \nabla f(z; W), B \right\rangle \right|.
$$

Upper bounding this expression by taking the supremum separately over each of terms,

$$
\sup_{\substack{\|x\|\le 1 \\ \|V-W\|\le R_V \\ \|B-W\|\le R_B}} \left|\left\langle \nabla f(x;V) - \nabla f(z;V), B\right\rangle\right| + \sup_{\substack{\|x\|\le 1 \\ \|V-W\|\le R_V \\ \|B-W\|\le R_B}} \left|\left\langle \nabla f(x;W) - \nabla f(z;W), B\right\rangle\right|
$$

$$
+ \sup_{\substack{\|x\|\le 1 \\ \|V-W\|\le R_V \\ \|B-W\|\le R_B}} \left|\left\langle \nabla f(z;V) - \nabla f(z;W), B\right\rangle\right|,
$$

and noticing the second term is bounded above by the first term, results in

$$
\sup_{\substack{\|x\|\le 1 \\ \|V-W\|\le R_V \\ \|B-W\|\le R_B}} \left|\left\langle \nabla f(x;V) - \nabla f(x;W), B\right\rangle\right|
$$

$$
\le \sup_{\substack{\|x\|\le 1 \\ \|V-W\|\le R_V \\ \|B-W\|\le R_B}} 2\left|\left\langle \nabla f(x;V) - \nabla f(z;V), B\right\rangle\right| + \sup_{\substack{\|x\|\le 1 \\ \|V-W\|\le R_V \\ \|B-W\|\le R_B}} \left|\left\langle \nabla f(z;V) - \nabla f(z;W), B\right\rangle\right|
$$

$$
\le \sup_{\substack{\|x\|\le 1 \\ \|V-W\|\le R_V \\ \|B-W\|\le R_B}} 2\left|\left\langle \nabla f(x;V) - \nabla f(z;V), B-V\right\rangle\right| + 2\left|f(x;V) - f(z;V)\right|
$$

$$
+ \sup_{\substack{\|x\|\le 1 \\ \|V-W\|\le R_V \\ \|B-W\|\le R_B}} \left|\left\langle \nabla f(z;V) - \nabla f(z;W), B\right\rangle\right|
$$

$$
\le \sup_{\substack{\|x\|\le 1 \\ \|V-W\|\le R_V \\ \|B-W\|\le R_B}} 2\|\nabla f(x;V) - \nabla f(z;V)\|(R_B + R_V) + 2\left|f(x;V) - f(z;V)\right|
$$

$$
+ \sup_{\substack{\|x\|\le 1 \\ \|V-W\|\le R_V \\ \|B-W\|\le R_B}} \left|\left\langle \nabla f(z;V) - \nabla f(z;W), B\right\rangle\right|.
$$

Instantiating Lemma A.7 part 1 of Ji et al. (2021) for all $z \in \mathcal{C}$, we get that with probability at least $1 - 3(\sqrt{d}/\epsilon)^d\delta$,

$$
\left|\left\langle \nabla f(z;V) - \nabla f(z;W), B\right\rangle\right| \le \frac{3\rho(R_B + 2R_V)R_V^{1/3}\ln(e/\delta)^{1/4}}{m^{1/6}}.
$$

By Lemma A.2 part 2 of Ji et al. (2021), with probability at least $1 - (\sqrt{d}/\epsilon)^d\delta$, $\|W\| \le (\sqrt{m} + \sqrt{d} + \sqrt{2\ln(1/((\sqrt{d}/\epsilon)^d\delta))})$. Assuming this holds, then for all $\|x - z\| \le \epsilon$,

$$
\left|f(x;V) - f(z;V)\right| = \left|\frac{\rho}{\sqrt{m}}\sum_{j=1}^{m} a_i\left(\sigma(v_j^\mathsf{T}x) - \sigma(v_j^\mathsf{T}z)\right)\right| \le \frac{\rho}{\sqrt{m}}\sum_{j=1}^{m}\left|\sigma(v_j^\mathsf{T}x) - \sigma(v_j^\mathsf{T}z)\right|
$$

$$
\le \frac{\rho}{\sqrt{m}}\sum_{j=1}^{m}\left|v_j^\mathsf{T}x - v_j^\mathsf{T}z\right| \le \frac{\rho}{\sqrt{m}}\sqrt{m}\|V\|\|x - z\| \le \rho(R_V + \|W\|)\epsilon
$$

$$
\le \rho(R_V + \sqrt{m} + \sqrt{d} + \sqrt{2\ln(1/((\sqrt{d}/\epsilon)^d\delta))})\epsilon.
$$

Instantiating Lemma A.12 for all $z \in \mathcal{C}$, we get that with probability at least $1 - (\sqrt{d}/\epsilon)^d\delta$,

$$
\sup_{\substack{\|x\|\le 1 \\ \|V-W\|\le R_V}} \|\nabla f(x;V) - \nabla f(z;V)\|
$$

$$
\le 7\rho R_V^{1/3}m^{-1/6}\left(\ln(em/\delta)\right)^{1/6} + 12\rho d^{1/6}\epsilon^{1/3}\left(\ln(em/\delta)\right)^{1/3} + 2\rho\epsilon
$$

$$
+ 15\rho\left(\frac{\ln(edm/\delta)}{m}\right)^{1/4}.
$$

Altogether, with probability at least $1 - 5(\sqrt{d}/\epsilon)^d \delta$,

$$
\sup_{\substack{\|x\| \leq 1 \\ \|V - W\| \leq R_V \\ \|B - W\| \leq R_B}} \left| \langle \nabla f(x; V) - \nabla f(x; W), B \rangle \right|
$$

$$
\leq 2 \Bigg( 7\rho R_V^{1/3} m^{-1/6} \left( \ln(em/\delta) \right)^{1/6} + 12\rho d^{1/6} \epsilon^{1/3} \left( \ln(em/\delta) \right)^{1/3} + 2\rho\epsilon
$$

$$
+ 15\rho \left( \frac{\ln(edm/\delta)}{m} \right)^{1/4} \Bigg) (R_B + R_V)
$$

$$
+ 2\rho(R_V + \sqrt{m} + \sqrt{d} + \sqrt{2\ln(1/((\sqrt{d}/\epsilon)^d \delta))})\epsilon
$$

$$
+ \frac{3\rho(R_B + 2R_V) R_V^{1/3} \ln(e/\delta)^{1/4}}{m^{1/6}}.
$$

Setting $\epsilon = 1/(dm)$ we get with probability at least $1 - 5(d\sqrt{d}m)^d \delta$,

$$
\sup_{\substack{\|x\| \leq 1 \\ \|V - W\| \leq R_V \\ \|B - W\| \leq R_B}} \left| \langle \nabla f(x; V) - \nabla f(x; W), B \rangle \right| \leq \frac{20\rho(R_B + R_V) R_V^{1/3} \ln(em/\delta)^{1/4}}{m^{1/6}}
$$

$$
+ \frac{64\rho(R_B + R_V) \ln(edm/\delta)^{1/3}}{m^{1/4}}
$$

$$
+ \frac{2\sqrt{2}\rho\sqrt{\ln(1/((\sqrt{d}/\epsilon)^d \delta))}}{dm}.
$$

Rescaling $\delta$ we get that with probability at least $1 - \delta$,

$$
\sup_{\substack{\|x\| \leq 1 \\ \|V - W\| \leq R_V \\ \|B - W\| \leq R_B}} \left| \langle \nabla f(x; V) - \nabla f(x; W), B \rangle \right| \leq \frac{20\rho(R_B + R_V) R_V^{1/3} d^{1/4} \ln(5ed^2 m^2/\delta)^{1/4}}{m^{1/6}}
$$

$$
+ \frac{64\rho(R_B + R_V) d^{1/3} \ln(5ed^3 m^2/\delta)^{1/3}}{m^{1/4}}
$$

$$
+ \frac{2\sqrt{2}\rho\sqrt{\ln(5/\delta)}}{dm}
$$

$$
\leq \frac{26\rho(R_B + R_V) R_V^{1/3} d^{1/4} \ln(ed^2 m^2/\delta)^{1/4}}{m^{1/6}}
$$

$$
+ \frac{89\rho(R_B + R_V) d^{1/3} \ln(ed^3 m^2/\delta)^{1/3}}{m^{1/4}}
$$

$$
+ \frac{5\rho\sqrt{\ln(1/\delta)}}{dm}.
$$

2. With probability at least $1 - \delta$, the previous part holds. This part is then an immediate consequence since the logistic loss is 1-Lipschitz.

$\square$

We now prove the main optimization lemma.

*Proof of Lemma 4.6.* We will use the fact that it does not matter much which feature we use, because the function values on the domain will differ by a small amount, and hence the resulting difference in risk will be small. This is encapsulated by Lemma A.13.

We stop training once we reach $t$ iterations, or the parameter distance from initialization exceeds $2R_Z$. That is, we stop at iteration $T$, where $T = \min\{t, \inf\{i : \|W_i - W_0\| > 2R_Z\}\}$. By definition $\|W_i - W_0\| \le 2R_Z \le R_{\mathrm{gd}}$ for all $i < T$, and

$$\|W_T - W_0\| \le \|W_{T-1} - W_0\| + \eta\|\nabla\widehat{\mathcal{R}}_{\mathrm{A}}(W_{T-1})\| \le 2R_Z + \eta\rho = R_{\mathrm{gd}}.$$

Then by Lemma A.13 part 2, we have that with probability at least $1 - \delta$,

$$\sup_{\substack{\|W_i - W\| \le R_{\mathrm{gd}} \\ \|B - W\| \le R_{\mathrm{gd}}}} \left|\widehat{\mathcal{R}}_{\mathrm{A}}^{(i)}(B) - \widehat{\mathcal{R}}_{\mathrm{A}}^{(0)}(B)\right|$$

$$\le \frac{52\rho R_{\mathrm{gd}}^{4/3} d^{1/4} \ln(ed^2 m^2/\delta)^{1/4}}{m^{1/6}} + \frac{178\rho R_{\mathrm{gd}} d^{1/3} \ln(ed^3 m^2/\delta)^{1/3}}{m^{1/4}} + \frac{5\rho\sqrt{\ln(1/\delta)}}{dm} := \kappa_1.$$

Note that this holds for all iterations of interest as well as when $B$ represents the reference parameters.

By Lemma A.11 and the interchangeability of features, we get

$$\|W_T - Z\|^2 + (2\eta - \eta^2\rho^2)\sum_{i<T}\widehat{\mathcal{R}}_{\mathrm{A}}(W_i) \le \|W_0 - Z\|^2 + 2\eta\sum_{i<T}\widehat{\mathcal{R}}_{\mathrm{A}}^{(i)}(Z)$$

$$\le \|W_0 - Z\|^2 + 2\eta\sum_{i<T}\left(\widehat{\mathcal{R}}_{\mathrm{A}}^{(0)}(Z) + \kappa_1\right)$$

$$= \|W_0 - Z\|^2 + 2\eta T\widehat{\mathcal{R}}_{\mathrm{A}}^{(0)}(Z) + 2\eta T\kappa_1.$$

Rearranging and using the definition of $W_{\le t}$ gives

$$\widehat{\mathcal{R}}_{\mathrm{A}}(W_{\le t}) \le \frac{1}{T}\sum_{i<T}\widehat{\mathcal{R}}_{\mathrm{A}}(W_i) \le \frac{2}{2 - \eta\rho^2}\widehat{\mathcal{R}}_{\mathrm{A}}^{(0)}(Z) + \frac{2}{2 - \eta\rho^2}\kappa_1 + \frac{\|W_0 - Z\|^2 - \|W_T - Z\|^2}{T(2\eta - \eta^2\rho^2)}.$$

It remains to bound the term $\frac{\|W_0 - Z\|^2 - \|W_T - Z\|^2}{T(2\eta - \eta^2\rho^2)}$. Note that if $\|W_T - Z\| \ge \|W_0 - Z\|$, we can bound the term above by $0$. Otherwise, we have

$$\|W_T - W_0\| \le \|W_T - Z\| + \|Z - W_0\| < 2\|W_0 - Z\| \le 2R_Z,$$

so we must have $T = t$. Using this bound results in

$$\frac{\|W_0 - Z\|^2 - \|W_T - Z\|^2}{T(2\eta - \eta^2\rho^2)} \le \frac{R_Z^2}{t(2\eta - \eta^2\rho^2)},$$

giving us the final bound. $\qquad\square$

## A.4 Adversarial Training Results

We can now prove our results on adversarial training.

*Proof of Theorem 4.1.* To bound the risk of our final iterate, we will first linearize it, apply our generalization bound to get the linearized training risk, use our optimization lemma to get the linearized training risk of the reference model, and then repeat our steps in reverse to get the risk of the linearized finite reference model. Let

$$\kappa_1 := \frac{52\rho R_{\mathrm{gd}}^{4/3} d^{1/4} \ln(ed^2 m^2/\delta)^{1/4}}{m^{1/6}} + \frac{178\rho R_{\mathrm{gd}} d^{1/3} \ln(ed^3 m^2/\delta)^{1/3}}{m^{1/4}} + \frac{5\rho\sqrt{\ln(1/\delta)}}{dm},$$

$$\kappa_n := \frac{2}{\sqrt{n}} + \frac{2\tilde{\tau}}{\sqrt{n}}\left(1 + \sqrt{m\ln\left(\frac{mn}{\tilde{\tau}^2}\right)}\right) + \frac{77d\ln^{3/2}(4em^2 d^3/\delta)}{\sqrt{n}}.$$

By Lemma A.13, with probability at least $1 - \delta$, we have $\mathcal{R}_{\mathrm{A}}(W_{\le t}) \le \mathcal{R}_{\mathrm{A}}^{(0)}(W_{\le t}) + \kappa_1$ and $\widehat{\mathcal{R}}_{\mathrm{A}}^{(0)}(W_{\le t}) \le \widehat{\mathcal{R}}_{\mathrm{A}}(W_{\le t}) + \kappa_1$. By Lemma 4.6, with probability at least $1 - \delta$, we have

$$\widehat{\mathcal{R}}_{\mathrm{A}}(W_{\le t}) \le \frac{2}{2 - \eta\rho^2}\widehat{\mathcal{R}}_{\mathrm{A}}^{(0)}(Z) + \frac{R_Z^2}{t(2\eta - \eta^2\rho^2)} + \frac{2}{2 - \eta\rho^2}\kappa_1.$$

By Lemma 4.4, with probability at least $1 - 5\delta$, we have $\mathcal{R}_{\mathrm{A}}^{(0)}(W_{\leq t}) \leq \widehat{\mathcal{R}}_{\mathrm{A}}^{(0)}(W_{\leq t}) + \rho R_{\mathrm{gd}}\kappa_n$, and with another probability at least $1 - 5\delta$ we have $\widehat{\mathcal{R}}_{\mathrm{A}}^{(0)}(Z) \leq \mathcal{R}_{\mathrm{A}}^{(0)}(Z) + \rho R_Z \kappa_n$. Adding several of these inequalities together,

$$\mathcal{R}_{\mathrm{A}}(W_{\leq t}) \leq \mathcal{R}_{\mathrm{A}}^{(0)}(W_{\leq t}) + \kappa_1,$$
$$\mathcal{R}_{\mathrm{A}}^{(0)}(W_{\leq t}) \leq \widehat{\mathcal{R}}_{\mathrm{A}}^{(0)}(W_{\leq t}) + \rho R_{\mathrm{gd}}\kappa_n,$$
$$\widehat{\mathcal{R}}_{\mathrm{A}}^{(0)}(W_{\leq t}) \leq \widehat{\mathcal{R}}_{\mathrm{A}}(W_{\leq t}) + \kappa_1,$$
$$\widehat{\mathcal{R}}_{\mathrm{A}}(W_{\leq t}) \leq \frac{2}{2 - \eta\rho^2}\widehat{\mathcal{R}}_{\mathrm{A}}^{(0)}(Z) + \frac{R_Z^2}{t(2\eta - \eta^2\rho^2)} + \frac{2}{2 - \eta\rho^2}\kappa_1,$$

and cancelling results in

$$\mathcal{R}_{\mathrm{A}}(W_{\leq t}) \leq \frac{2}{2 - \eta\rho^2}\widehat{\mathcal{R}}_{\mathrm{A}}^{(0)}(Z) + \frac{R_Z^2}{t(2\eta - \eta^2\rho^2)} + \rho R_{\mathrm{gd}}\kappa_n + \left(2 + \frac{2}{2 - \eta\rho^2}\right)\kappa_1.$$

Using $\widehat{\mathcal{R}}_{\mathrm{A}}^{(0)}(Z) \leq \mathcal{R}_{\mathrm{A}}^{(0)}(Z) + \rho R_Z \kappa_n$ and simplifying gives the final bound, as follows.

$$\mathcal{R}_{\mathrm{A}}(W_{\leq t})$$
$$\leq \frac{2}{2 - \eta\rho^2}\mathcal{R}_{\mathrm{A}}^{(0)}(Z) + \frac{R_Z^2}{t(2\eta - \eta^2\rho^2)} + \rho\left(R_{\mathrm{gd}} + \frac{2}{2 - \eta\rho^2}R_Z\right)\kappa_n + \left(2 + \frac{2}{2 - \eta\rho^2}\right)\kappa_1$$
$$\leq \frac{2}{2 - \eta\rho^2}\mathcal{R}_{\mathrm{A}}^{(0)}(Z) + \frac{R_Z^2}{t(2\eta - \eta^2\rho^2)} + \rho\left(2R_Z + \eta\rho + \frac{2}{2 - \eta\rho^2}R_Z\right)\widetilde{\mathcal{O}}\left(\frac{d + \sqrt{\tau m}}{\sqrt{n}}\right)$$
$$\quad + \rho\left(2 + \frac{2}{2 - \eta\rho^2}\right)\widetilde{\mathcal{O}}\left(\frac{(2R_Z + \eta\rho)^{4/3}d^{1/4}}{m^{1/6}} + \frac{(2R_Z + \eta\rho)d^{1/3}}{m^{1/4}} + \frac{1}{dm}\right)$$
$$\leq \frac{2}{2 - \eta\rho^2}\mathcal{R}_{\mathrm{A}}^{(0)}(Z) + \widetilde{\mathcal{O}}\left(\left(\frac{1}{2 - \eta\rho^2}\right)\left(\frac{R_Z^2}{\eta t} + \frac{\rho R_Z\left(d + \sqrt{\tau m}\right)}{\sqrt{n}} + \frac{\rho R_Z^{4/3}d^{1/3}}{m^{1/6}}\right)\right).$$

$\square$

Setting parameters appropriately, we can make all terms in Theorem 4.1 small, and get arbitrarily close to the optimal adversarial convex loss. This is encapsulated by Corollaries 4.2 and 4.3, which we prove at the same time.

*Proof of Corollary 4.2 and Corollary 4.3.* By definition and by Lemma 3.4, we can find a continuous function $h$ such that

$$\mathcal{R}_{\mathrm{A}}(h) \leq \inf\{\mathcal{R}_{\mathrm{A}}(g) : g \text{ continuous}\} + \epsilon/2 = \inf\{\mathcal{R}_{\mathrm{A}}(g) : g \text{ measurable}\} + \epsilon/2.$$

By Theorem 4.3 of Ji et al. (2019), we can find an infinite-width network $|f(\frac{1}{\sqrt{2}}(x; 1); U_\infty^\epsilon) - h(\frac{1}{\sqrt{2}}(x; 1))| \leq \epsilon/2$, with associated $R_\epsilon := \max\{\rho, \eta\rho^2, \sup_x \|U_\infty^\epsilon(x)\|\} < \infty$. Define $\kappa_2 := 6\rho d \ln(emd^2/\delta) + \frac{20R\sqrt{d\ln(ed^3m^2/\delta)}}{m^{1/4}}$. Then within the proof of Lemma A.11 of Ji et al. (2021) it is shown that with probability at least $1 - 6\delta$, we can sample finite width reference parameters $Z$ such that $|f^{(0)}(\frac{1}{\sqrt{2}}(x; 1); Z) - f(\frac{1}{\sqrt{2}}(x; 1); U_\infty^\epsilon)| \leq \kappa_2$. By the triangle inequality, $|f^{(0)}(\frac{1}{\sqrt{2}}(x; 1); Z) - h(\frac{1}{\sqrt{2}}(x; 1))| \leq \kappa_2 + \epsilon/2$ for all $\|x\| \leq 1$. So for any $\|x\| \leq 1$, $|\ell_{\mathrm{A}}(x, y, f(\frac{1}{\sqrt{2}}(x; 1); Z)) - \ell_{\mathrm{A}}(x, y, h(\frac{1}{\sqrt{2}}(x; 1)))| \leq \kappa_2 + \epsilon/2$. As a result,

$$\mathcal{R}_{\mathrm{A}}^{(0)}(Z) \leq \mathcal{R}_{\mathrm{A}}(h) + \kappa_2 + \epsilon/2 \leq \inf\{\mathcal{R}_{\mathrm{A}}(g) : g \text{ measurable}\} + \kappa_2 + \epsilon.$$

Combining this with Theorem 4.1 holding with probability at least $1 - 12\delta$, and so altogether with probability at least $1 - 18\delta$,

$$
\mathcal{R}_{\mathrm{A}}(W_{\leq t}) \leq \frac{2}{2 - \eta\rho^2} \mathcal{R}_{\mathrm{A}}^{(0)}(Z) + \widetilde{\mathcal{O}}\left( \left( \frac{1}{2 - \eta\rho^2} \right) \left( \frac{R_Z^2}{\eta t} + \frac{\rho R_Z \left( d + \sqrt{\tau m} \right)}{\sqrt{n}} + \frac{\rho R_Z^{4/3} d^{1/3}}{m^{1/6}} \right) \right)
$$

$$
\leq \frac{2}{2 - \eta\rho^2} \inf_{g \text{ meas.}} \{ \mathcal{R}_{\mathrm{A}}(g) \} + \frac{2}{2 - \eta\rho^2} (\kappa_2 + \epsilon)
$$

$$
+ \widetilde{\mathcal{O}}\left( \left( \frac{1}{2 - \eta\rho^2} \right) \left( \frac{R_Z^2}{\eta t} + \frac{\rho R_Z \left( d + \sqrt{\tau m} \right)}{\sqrt{n}} + \frac{\rho R_Z^{4/3} d^{1/3}}{m^{1/6}} \right) \right).
$$

Corollary 4.2 then follows by setting parameters and reducing.

To get Corollary 4.3, let $\delta^{(n)} = n^{-2}$. Notice that for any $\epsilon > 0$, there exists $n_\epsilon$ such that for all $n \geq n_\epsilon$, with probability at least $1 - 1/n^2$,

$$
\mathcal{R}_{\mathrm{A}}(W_{\leq t})^{(n)} \leq \inf_{g \text{ meas.}} \{ \mathcal{R}_{\mathrm{A}}(g) \} + \epsilon.
$$

Since $\sum_{n \geq n_\epsilon} 1/n^2 < \infty$, by the Borel-Cantelli lemma we have

$$
\limsup_{n \to \infty} \mathcal{R}_{\mathrm{A}}(W_{\leq t}^{(n)}) \leq \inf_{g \text{ meas.}} \{ \mathcal{R}_{\mathrm{A}}(g) \} + \epsilon
$$

almost surely.

As $\epsilon > 0$ was arbitrary, we get Corollary 4.3. $\qquad\square$

