# OpenReview forum: "On Achieving Optimal Adversarial Test Error"
_ICLR.cc/2023/Conference — ICLR 2023 poster_

### Official Review · Reviewer_k1gN · 2022-10-24

**Confidence:** 3
**Correctness:** 3
**Technical Novelty And Significance:** 3
**Empirical Novelty And Significance:** Not applicable
**Recommendation:** 6

**Clarity, Quality, Novelty And Reproducibility:**

Clarity: the paper delivers well on its core message. But some details need some future clarifications to avoid confusion.
Novelty: I believe the result is new, and generalizes existing results on clean training.
Reproducibility: N.A.

**Strength And Weaknesses:**

Strength: Overall the paper delivers reasonable theoretical results.
Weakness: results in section 4 are restricted to a relatively simple model; results in section 3 and section 4 are not well integrated, and they feels like two separate topics.

Thm 4.1: I don't understand the role of t. If we let t go to infinite, the definition of W_{<=t} essentially doesn't change. However, one of the terms in upper bound decays to zero.

Col 4.2: The condition says m needs to be bounded (big O notation). But I believe the author(s) means that m needs to be larger than the given rate.

The conditions of Lemma 4.4-4.6 are not mentioned in theorem 4.1.

According to Thm 3.3, to get a good zero-one loss, one can train a model based on convex loss and tune the threshold t value. Does Fig 1 reflect this tuning?

**Summary Of The Paper:**

This paper contains two parts. The first part discusses properties of optimal adversarial estimators w.r.t. convex loss and 0-1 loss, including their connection. The second part discusses adversarial lazy training of one-layer wide network, and demonstrates a convergence of population adversarial loss.

**Summary Of The Review:**

I currently recommend marginal acceptance.
Although the results are new, but it does depends on limited model setup.

---

> ### Author Response · Authors · 2022-11-17
> **Response to Reviewer k1gN**
>
> Thank you for the review and the helpful suggestions.
>
> ### Weaknesses
> >Weakness: results in section 4 are restricted to a relatively simple model
>
> The 2-layer ReLU network we consider is one of the simplest settings which preserves several nontrivial properties of modern architectures --- a nonconvex optimization problem, universal approximation, and nondifferentiability at some points of the network --- that we must deal with in our analysis.
> It would be interesting future work to extend our results to modern architectures/settings.
>
> >results in section 3 and section 4 are not well integrated, and they feels like two separate topics
>
> While the results are quite distinct between the two sections,
> we note that Lemma 3.4 is used within section 4,
> and Theorem 3.3 justifies the focus of section 4 being on minimizing the adversarial convex loss, as this also minimizes the adversarial zero-one loss (with proper thresholding).
>
> >Thm 4.1: I don't understand the role of t. If we let t go to infinite, the definition of W_{<=t} essentially doesn't change. However, one of the terms in upper bound decays to zero.
>
> The hyperparameter $t$ is the maximum number of iterations in our gradient descent algorithm.
> You are correct in that taking $t$ larger only helps our bound.
> However, setting $t$ to be finite ensures that the algorithm stops in a finite amount of time.
>
> >Col 4.2: The condition says m needs to be bounded (big O notation). But I believe the author(s) means that m needs to be larger than the given rate.
>
> Thank you, we have now fixed this.
>
> >The conditions of Lemma 4.4-4.6 are not mentioned in theorem 4.1.
>
> Thank you, we have now fixed this.
>
> >According to Thm 3.3, to get a good zero-one loss, one can train a model based on convex loss and tune the threshold t value. Does Fig 1 reflect this tuning?
>
> Fig 1 does not reflect this tuning. Other than using a constant step size it is a reproduction of prior experiments from [Rice et al., 2020].
> However, this is an interesting avenue for future work.
> * [Rice et al., 2020] Leslie Rice, Eric Wong, and J. Zico Kolter. Overfitting in adversarially robust deep learning. https://arxiv.org/abs/2002.11569
>
> Let us know if you have any further questions or suggestions, and thank you again for the review.

---

### Official Review · Reviewer_Quhe · 2022-10-26

**Confidence:** 4
**Correctness:** 4
**Technical Novelty And Significance:** 4
**Empirical Novelty And Significance:** Not applicable
**Recommendation:** 8

**Clarity, Quality, Novelty And Reproducibility:**

Clarity: This paper is great

Quality and novelty: To the extent of my knowledge, the results seem to be interesting for the ICLR audience.

Reproducibility: there are no experiments (except Fig 1 which corresponds to reproducing an experiment from Rice et al.)


**Strength And Weaknesses:**

## Strengths:

- The results seem novel
- The question of learning efficient, robust classifiers is a critical question. Establishing connections between generalization and robustness is an important problem.
- The paper is well written, and the authors made an effort to make the results accessible to a broad audience of experts.

## Weaknesses:

- This paper does not provide any experiments to test the gap between their theory and the practice. Since showing non-vacuous standard generalization bounds for NNs is not fully solved, it seems too much to expect it is the more challenging setting of adversarial test risk. However, I think the author could still propose some experimental exploration in the same vein as what they mention in the discussion. (more width helps, influence of the perturbation radius,...)


**Summary Of The Paper:**

This paper provides generalization bounds for the adversarial risk of two-layers neural networks trained with early stopping and an ideal adversary (i.e., an adversary that computes perfect adversarial attacks). They do so by providing new results on the Rademacher complexity of the outcome of adversarial attacks on linearized two-layers NNs close to initialization.

**Summary Of The Review:**

This paper is well written. The presentation of the contributions and the results is clear and the results seem significant (to the extent of my knowledge). It is a clear accept to me.

---

> ### Author Response · Authors · 2022-11-17
> **Response to Reviewer Quhe**
>
> Thank you for the review and the insightful comments.
>
> ### Weaknesses
> We appreciate the suggestion, and have added discussion on future experimental exploration.
>
> Let us know if you have any further questions or suggestions, and thank you again for the review.

---

### Official Review · Reviewer_7qcR · 2022-11-03

**Confidence:** 3
**Correctness:** 3
**Technical Novelty And Significance:** 3
**Empirical Novelty And Significance:** 2
**Recommendation:** 6

**Clarity, Quality, Novelty And Reproducibility:**

# Clarity
- Figure 1 seems misleading -- it implies that the paper tackles the "early phase" of training before robust overfitting kicks in, which could span on the order of tens of epochs. In contrast, if I understand correctly (according to e.g. Figure 1 in [2]), the "early phase" considered in this paper is significantly shorter -- empirically one would expect it to span only a few batch updates.


# Novelty
The proposed approach is a clever combination of the idea that only *calibrated* convex losses are useful as surrogates for the 0-1 loss in adversarially robust classification [1], and the ideas presented in [2] relating early stopping and calibration


[1] Bao, Han, Clay Scott, and Masashi Sugiyama. "Calibrated surrogate losses for adversarially robust classification." Conference on Learning Theory. PMLR, 2020.

[2] Ji, Ziwei, Justin Li, and Matus Telgarsky. "Early-stopped neural networks are consistent." Advances in Neural Information Processing Systems 34 (2021): 1805-1817.

**Strength And Weaknesses:**

# Strengths

The results connecting zero-one losses with convex surrogates are elegant and easy to follow. In particular, I like how the simple statement of Lemma 3.1 opens the door for the remaining results in Section 3.

# Weaknesses
There are multiple axes along which the current paper falls short of applying to realistic settings: 1) the assumption that one is given an oracle adversary, i.e. we have access to the worst-case perturbation (as opposed to a noisy gradient oracle, i.e. just doing PGD); 2) the results in section 4 apply only to shallow fully-connected ReLU networks; 3) the results hold only in a regime very close to initialization and it is assumed one has an early stopping criterion/oracle.

Weaknesses 2) and 3) are not unique to this work, and thus I heavily discount their severity when considering my overall recommendation.

**Summary Of The Paper:**

The authors consider two problems: 1) relating convex surrogate losses with zero-ones losses in adversarial training; 2) bounding the (adversarial) convex risk of shallow ReLU networks.

For the former, the authors' results apply to a very broad setting -- arbitrary perturbation sets and general data distributions. They show a global relation between convex and zero-one losses. Previous work indicated that convex losses may not be calibrated, making them inadequate surrogates for the zero-one loss in adversarial settings. The authors show that if an optimal threshold is chosen, one can bound the zero-one loss via the convex surrogate.

For the latter problem, the authors restrict their attention to $\ell_2$ perturbations. Following the approach of [2], the authors characterize the (adversarial) risk near initialization of training shallow ReLU networks using the logistic loss.

**Summary Of The Review:**

The authors present novel ideas about convex surrogates for adversarial training, and extend previous works on generalization from standard training to the adversarial training. The paper is well-written. Overall, I believe it will be a valuable addition to the field. Thus, I recommend acceptance.

---

> ### Author Response · Authors · 2022-11-17
> **Response to Reviewer 7qcR**
>
> Thank you for the review and the insightful comments.
>
> ### Weaknesses
> We agree that that all of the weaknesses mentioned are shortcomings, and resolving any combination of them would be interesting future work.
>
> ### Clarity
> You are correct that the "early phase" considered in this paper is significantly shorter.
> Our intention was to show we are within the phase where getting small generalization error is possible,
> not that we can analyze the entire regime where this is the case.
> We have edited the accompanying caption to Figure 1 to clarify this.
>
> Let us know if you have any further questions or suggestions, and thank you again for the review.

---

### Decision · Program_Chairs · 2023-01-20

**Decision:**

Accept: poster

**Justification For Why Not Higher Score:**

The results seem fine, but may be of limited interest.

**Justification For Why Not Lower Score:**

Clearly above the acceptance threshold.

**Metareview: Summary, Strengths And Weaknesses:**

The paper considers two problems. The first is relating adversarial zero-one loss to convex surrogates. The results in this section are clearly written and easy to follow; they also apply to a broad setting. The second problem is about adversarial lazy training of one-layer wide network. They restrict attention to l_2 loss and characterize the adversarial risk near initialization of training such networks using the logistic loss.



**Note From Pc:**

if the above contains the word "oral" or "spotlight" please see: "oral" presentation means -> notable-top-5% and "spotlight" means -> notable-top-25%. As stated in our emails, we are disassociating presentation type from AC recommendations